# PRISM: Perception Reasoning Interleaved for Sequential Decision Making

**Mohamed Salim Aissi** [1]  **Clemence Grislain** [1]  **Clement Romac** [2 3]  **Laure Soulier** [1]
**Mohamed Chetouani** [1]  **Olivier Sigaud** [1]  **Nicolas Thome** [1 4 5]

## Abstract

Scaling LLM-based embodied agents from text-only environments to complex multimodal settings remains a major challenge. Recent work identifies a perception–reasoning–decision gap in standalone Vision–Language Models (VLMs), which often overlook task-critical information. In this paper, we introduce PRISM, a framework that tightly couples perception (VLM) and decision (LLM) through a dynamic question–answer (DQA) pipeline. Instead of passively accepting the VLM's description, the LLM critiques it, probes the VLM with goal-oriented questions, and synthesizes a compact image description. This closed-loop interaction yields a sharp, task-driven understanding of the scene. We evaluate PRISM on the ALFWorld and Room-to-Room (R2R) benchmarks. We show that: (1) PRISM significantly outperforms state-of-the-art image-based models, (2) our Interactive goal-oriented perception pipeline yields systematic and substantial gains, and (3) PRISM is fully automatic, eliminating the need for handcrafted questions or answers.

## 1. Introduction

The field of Embodied Artificial Intelligence, which aims to develop agents capable of perceiving and autonomously interacting with the physical world, has attracted significant attention since the emergence of foundation models.

In particular, Large Language Models (LLMs) and Vision-Language Models (VLMs) have demonstrated remarkable emergent capabilities in logical reasoning, semantic understanding, and common-sense knowledge (Ichter et al., 2023; Zhai et al., 2024; Li et al., 2025). This naturally led to

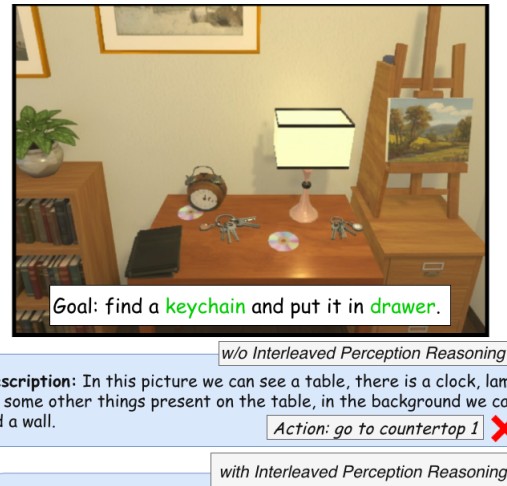

*Figure 1.* **Illustration of** PRISM**'s interleaved perception (VLM) and reasoning (LLM).** Given an image and goal, PRISM's dynamic question-answering (DQA) enables the LLM to query the VLM for task-critical information. This yields fine-grained, task-driven descriptions (bottom), ensuring accurate action prediction. Conversely, methods decoupling the VLM and LLM yield task-agnostic, incomplete descriptions, causing the agent to fail.

growing interest in studying how to integrate them as components in embodied agents that interact with the world. Our work studies how to incorporate them as components of agents interacting with visual environments in order to solve natural language instructions.

Prior research has shown that LLMs can be leveraged for sequential decision-making when provided with a textual description of the environment, either zero-shot (Yao et al., 2023; Shinn et al., 2023), or through fine-tuning (Carta et al., 2023; Wang et al., 2023). These approaches typically assume perfect perception, where prompts contain all necessary information. However, transitioning from such idealized text to raw visual input remains a major challenge.

[1]Sorbonne Universite, CNRS, ISIR, F-75005 Paris, France [2]Hugging Face [3]Inria (Flowers), University of Bordeaux, France [4]Institut Universitaire de France (IUF) [5]International Laboratory on Learning Systems (ILLS). Correspondence to: Mohamed Salim Aissi <Mohamed-Salim.aissi@Sorbonne-universite.fr>.

*Proceedings of the 43rd International Conference on Machine Learning*, Seoul, South Korea. PMLR 306, 2026. Copyright 2026 by the author(s).

The impressive ability of VLMs to process images and generate text naturally positions them as candidates for this challenge (Alayrac et al., 2022; Liu et al., 2023). Recent works have explored two primary integration strategies: (1) directly replacing the LLM with a VLM (Zhai et al., 2024; Yang et al., 2024), or (2) employing the VLM as an intermediate perception module to generate textual descriptions for a decision-making LLM (Pan et al., 2024; Zhou et al., 2023; Aissi et al., 2025). However, performance still lags behind that achieved in textual environments. For the latter, this gap highlights the limited ability of VLMs to capture task-critical, fine-grained visual information when perception (VLM) and decision-making (LLM) are loosely coupled.

In contrast, perception and decision-making are deeply intertwined in humans, where sensory inputs prioritize information essential to the current goal (Gibson, 1986). To mirror this interactivity, recent works have proposed introducing feedback mechanisms from decision-making to perception, using human clarification or question-answering (Gao et al., 2022; Shen et al., 2024; Long et al., 2023). However, the integration of perception and reasoning for decision-making is achieved through naive solutions, *e.g.,* mere concatenation of textual interactions. More importantly, these approaches rely on predefined questions or answers, restricting their applicability to realistic settings.

In this paper, we introduce PRISM: **P**erception and **R**easoning **I**nterleaved for **S**equential Decision-**M**aking. PRISM establishes a tight coupling between perception and decision-making through a *dynamic question-answering* (DQA) mechanism. The perception module (VLM) provides an initial scene description, which the decision module (LLM) critiques against the goal to identify and query missing information. Once the VLM responds, the LLM synthesizes the interaction feedback into a compact, task-driven description. As illustrated in Figure 1, PRISM's interleaved reasoning yields superior task-oriented descriptions compared to approaches decoupling VLM and LLM modules (Zhou et al., 2023; Pan et al., 2024; Aissi et al., 2025).

**Our contributions can be summarized as follows:**

• **Dynamic Question-Answering (DQA):** DQA tightly couples perception and decision-making, and synthesizes interaction feedback into a compact textual scene description. Crucially, PRISM 's DQA is fully automatic, eliminating the need for unrealistic handcrafted or oracle annotations.

• **Efficient training with PRISM's architecture:** Separating perception via DQA from decision via the LLM enables the use of established recipes for text-based Reinforcement Learning (RL), while effectively grounding decisions through online interactions within the visual environment.

• **Extensive experiments** on ALFWorld (Shridhar et al., 2021) and Room-to-Room (R2R) (Anderson et al., 2018) show that PRISM significantly outperforms current image-based models. We demonstrate consistent gains from DQA

across both benchmarks and provide rich analyses validating the generated questions, answers, and descriptions.

## 2. Related Work

### 2.1. Visual sequential decision-making

To reach goals from visual inputs, VLM-based agents have been explored. As zero-shot generalist VLMs often lack sufficient grounding, recent research has shifted toward RL fine-tuning, *e.g.,* RL4VLM Zhai et al. (2024). However, these agents significantly underperform LLMs provided with perfect textual descriptions Yang et al. (2024), as discussed in introduction and demonstrated in our experiments (Section 4). This disparity highlights the limited decision-making capacity of current VLMs and the difficulty of directly applying vanilla RL—highly effective for text-based inputs Carta et al. (2023); Wang et al. (2023)—to visual sequential decision-making.

Further efforts have been made to design architectures combining VLMs and LLMs to leverage their complementary strengths. Typically, the VLM describes the visual scene while the LLM selects actions. Such systems have shown promise in navigation (Anderson et al., 2018; Zhou et al., 2023; Pan et al., 2024; Zhang et al., 2025) and interactive environments, *e.g.,* VIPER (Aissi et al., 2025). However, perception and decision-making remain largely decoupled in these approaches: the VLM operates independently of the agent's reasoning, often overlooking task-relevant visual details (see Figure 1). We adopt this modular architecture with a VLM for perception and an LLM for action selection, but introduce a dynamic question-answering (DQA) mechanism that tightly couples perception and decision-making.

### 2.2. Interactive question-answering in embodied agents

Asking questions is a powerful mechanism to acquire missing information (Testoni and Fernández, 2024), enabling agents to refine their environmental understanding for effective sequential decision-making. Prior work enabled agents to query external sources such as humans or oracles as in DialFRED (Gao et al., 2022) or (Nguyen et al., 2022; Chen et al., 2023), but typically relies on predefined questions and access to accurate oracles, limiting adaptability and autonomy in real-world settings.

Recent works such as Long et al. (2023) and DiscussNAV (Shen et al., 2024) allow agents to query VLM-based perception modules. However, these systems still rely on fixed, predefined questions. Furthermore, in DiscussNAV, answers are merely concatenated with scene descriptions, creating long, noisy prompts that hinder decision-making. In contrast, PRISM leverages the LLM to synthesize DQA interactions into a compact, task-driven prompt suitable for fine-tuning, including with RL. We demonstrate that PRISM outperforms

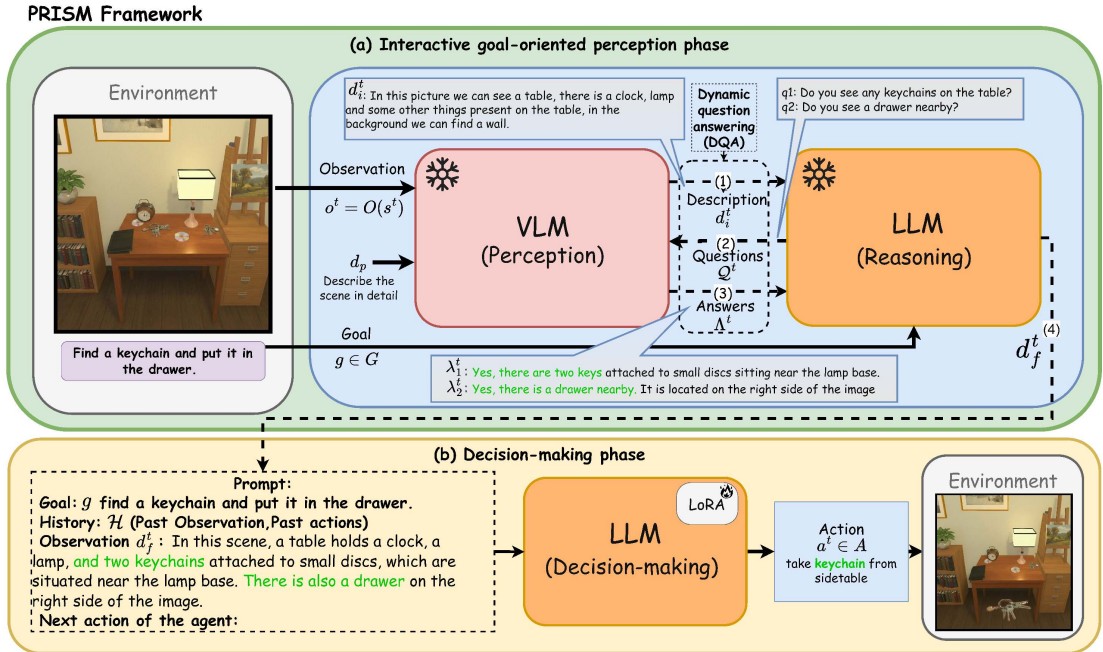

*Figure 2.* **The PRISM Framework:** The agent processes a goal $g$ and image observation $o^t$ to execute an action $a^t$. **(a) Interactive Goal-Oriented Perception:** A VLM and LLM collaborate **through dynamic question answering (DQA)** to refine scene understanding: **(1)** the VLM provides an initial description $d_i^t$; **(2)** the LLM generates goal-relevant questions $\mathcal{Q}^t$; **(3)** the VLM provides a specific answer $\lambda_i^t$ for each question $q_i^t \in \mathcal{Q}^t$; and **(4)** the LLM synthesizes these into a final description $d_f^t$. **(b) Decision-Making:** The same LLM, augmented with LoRA adapters trained for action generation, takes $d_f^t$, the goal $g$, and the history $\mathcal{H}$ to generate the action $a^t$ executed in the environment.

DiscussNAV in R2R with a simpler architecture and without relying on oracle questions or answers.

## 3. Method

### 3.1. Problem Statement

We consider a goal-conditioned partially observable Markov decision process $M = (S, A, T, R, G, O, \gamma)$, with state space $S$, deterministic transition function $T : S \times A \mapsto S$, goal-conditioned reward $R : S \times G \mapsto \mathbb{R}$, and discount factor $\gamma \in [0, 1)$. We assume a multimodal setting where actions $a^t \in A$ and goals $g \in G$ are textual commands and instructions. Given a language vocabulary $\mathcal{V}$ and a maximum sequence length $N$, we have $A$ and $G \in \mathcal{V}^N$. The state observation $o^t = O(s^t)$ is an RGB image in $\mathbb{R}^{3 \times H \times W}$, where $H \times W$ are image dimensions. Given an initial visual observation $o^0$ and a goal $g$, we want to find the policy $\pi_\theta : \mathbb{R}^{3 \times H \times W} \times G \mapsto A$ that maximizes the discounted sum of rewards obtained along its trajectory.

### 3.2. PRISM architecture

We propose PRISM (see Figure 2), an architecture integrating a VLM and an LLM across two distinct phases: *interactive goal-oriented perception* and *decision-making*.

During the interactive goal-oriented perception phase, the system generates an intermediate textual scene representation $d_f^t$. We design an interaction loop leveraging pretrained VLM and LLM knowledge to obtain $d_f^t$ with the necessary information for decision-making.

More precisely, to alleviate VLMs' limited ability to provide descriptions for decision-making (Zhang et al., 2025), we introduce an iterative communication process in which the LLM identifies informational gaps in the VLM initial description given the goal, and queries the VLM for specific details. The LLM then synthesizes the responses to update the environmental context, producing a refined final representation $d_f^t$ after the feedback interaction steps.

For decision-making, the same LLM used in the previous step acts as the policy, augmented with LoRA adapter (Hu et al., 2021) trained for action generation. Given the description $d_f^t$, goal, and interaction history, this module generates the next action $a^t \in \mathcal{A}$ to achieve the goal $g$.

**Interactive goal-oriented perception with VLM and LLM:** Standard modular perception-decision approaches that strictly decouple perception from reasoning, such as the ones proposed by Pan et al. (2024); Aissi et al. (2025) often produce generic scene descriptions with missing key information (see Figure 2(a)). Conversely, simply pro-

viding the goal to the perception module introduces a high risk of hallucinations, which significantly deteriorates downstream performance, as demonstrated in our experiments (see Section 4.4). To overcome this, we introduce a Dynamic Question-Answering (DQA) phase between the VLM, acting as the perception module ($\mathcal{V}_\mathcal{P}$) and the LLM, serving as the reasoning module ($\mathcal{R}$), to retrieve missing, task-critical information. At each time step $t$, the agent receives a visual observation $o^t$. The perception module $\mathcal{V}_p$ first produces an initial description $d_i^t$ by being prompted with a fixed goal-agnostic instruction $d_p$, specifically $d_p = $ "*Describe the scene in detail*":

$$d_i^t = \mathcal{V}_p(d_p, o^t). \tag{1}$$

The reasoning module $\mathcal{R}$ generate a set of questions $\mathcal{Q}^t$ adapted to bridge the information gap between the requirements of goal $g$ and the initial description $d_i^t$:

$$\mathcal{Q}^t = \{q_1^t, \dots, q_n^t\} = \mathcal{R}(d_i^t, g). \tag{2}$$

The perception module $\mathcal{V}_p$ is then queried with each question $q_j^t \in \mathcal{Q}^t$ to extract additional details from the current observation $o^t$. Then, the VLM directly produces a corresponding set of answers $\Lambda^t$:

$$\Lambda^t = \lambda_{j\ j=1}^{t\ |\mathcal{Q}^t|}, \qquad \lambda_j^t = \mathcal{V}_p(q_j^t, o^t). \tag{3}$$

This mechanism ensures a fully automated pipeline by removing the human from the perception loop. The performance of the agent with VLM answers are further evaluated in Section 4.3.3. Finally, rather than simply concatenating the raw QA pairs, as in (Long et al., 2023)—which results in an overlong and noisy input—we perform a semantic synthesis. The reasoning module $\mathcal{R}$ integrates $(d_i^t, \mathcal{Q}^t, \Lambda^t)$ to produce a refined description $d_f^t$. This approach leverages the LLM's summarization capabilities to distill the visual feedback into a concise representation, filtering out irrelevant details while emphasizing goal-critical elements, as shown in Figure 2(b), that is

$$d_f^t = \mathcal{R}(d_i^t, \mathcal{Q}^t, \Lambda^t). \tag{4}$$

The final description $d_f^t$ is then used for decision-making to generate appropriate actions $a^t \in A$ to achieve the goal $g$. Specific prompt templates for question generation, perception answers and final synthesis are detailed in Appendix A.

**Sequential decision-making with an LLM:** Building on the framework established by Carta et al. (2023), we define an LLM-based policy $\pi$ that directly maps the target goal $g$, the history of past transitions $\mathcal{H}$, and the refined textual description $d_f^{(t)}$ to an action $a^t \in \mathcal{A}$ (see Figure 2(b)). We compute an action probability $\pi_\theta(a_i|g, \mathcal{H}, d_f^t)$ as the probability assigned by the LLM of the action tokens

$a_i = \{w_0, \dots, w_{|a_i|}\}$ to follow the goal $g$, the history of past transitions $\mathcal{H}$ and the final description $d_f^t$:

$$\pi_\theta(a_i|g, \mathcal{H}, d_f^t) = \prod_{j=0}^{|a_i|} P_{LLM}(w_j|g, \mathcal{H}, d_f^t, w_{<j}). \tag{5}$$

The most probable action is then executed in the environment, which provides the next state and observation $s^{t+1}$ and $o^{t+1}$. In practice, the interactive goal-oriented perception and the decision-making phases share the same LLM backbone. However, the LoRA adapters are specifically trained for action generation and are only activated during the decision-making phase. The overall pipeline is summarized in Algorithm 1.

---

**Algorithm 1** PRISM

**Require:** Environment with observations $o^t$
    Task goal $g$
    Perception module $\mathcal{V}_p$
    LLM reasoning module $\mathcal{R}$
    Policy LLM $\pi_\theta$.
1: Initialize interaction history $\mathcal{H} \leftarrow \emptyset$
2: **for** each time step $t = 1, 2, \dots$ **do**
3:    $o^t \leftarrow O(s^t)$
4:    **Interactive goal-oriented perception phase**
5:    $d_i^t \leftarrow \mathcal{V}_p(d_p, o^t)$
6:    $\mathcal{Q}^t \leftarrow \mathcal{R}(d_i^t, g)$
7:    $\Lambda^t \leftarrow \{\lambda_j^t = \mathcal{V}_p(q_j^t, o^t)\}_{j=1}^{|\mathcal{Q}^t|}$
8:    $d_f^t \leftarrow \mathcal{R}(d_i^t, \mathcal{Q}^t, \Lambda^t)$
9:    **Sequential decision-making phase**
10:    $a^t \sim \pi_\theta(\cdot \mid g, \mathcal{H}, d_f^t)$
11:    Execute $a^t$
12:    Update interaction history: $\mathcal{H} \leftarrow \mathcal{H} \cup \{d_f^t, a^t\}$
13: **end for**

---

### 3.3. Training Strategies

Recent studies suggest that despite their vast common-sense knowledge, LLMs require fine-tuning on downstream tasks to align their linguistic representations with specific environmental constraints when used as policy (Carta et al., 2023; Zhai et al., 2024). We therefore fine-tune the LLM-based policy (with LoRA adapters) with the following two-stage training pipeline: pre-training via Behavioral Cloning (BC) followed by online Reinforcement Learning (RL).

Unlike methods that rely on data or supervision from parallel textual environments (Yang et al., 2024; Ao et al., 2025), our approach operates exclusively within the visual environment. Let $\mathcal{D} = \{o^t, \mathcal{H}, g, a^t, r^t\}$ be a demonstration dataset collected in a multimodal visual environment. We generate a textual version of this corpus, denoted as $\mathcal{D}_\text{text} = \{d_f^t, \mathcal{H}, g, a^t, r^t\}$, by substituting the initial visual

*Table 1.* **Success rate (SR) in ALFWorld:** Results marked with * are our reimplementation. The highest score for each task within a category is highlighted in bold. PRISM significantly outperforms methods with no access to the text environment and demonstrates more stable performance across all tasks. Notably, it reduces the gap to methods using textual supervision with privileged information.

| Supervision | Method | Agent type | Pick↑ | Look↑ | Clean↑ | Heat↑ | Cool↑ | Pick2↑ | Avg ↑ |
|---|---|---|---|---|---|---|---|---|---|
| Access to text env | ReAct | LLM | 71% | 28% | 65% | 62% | 44% | 35% | 54% |
| | DEPS | LLM | 93% | 100% | 50% | 80% | **100%** | 00% | 76% |
| | Reflexion | LLM | **96%** | **94%** | **100%** | 81% | 83% | 88% | **91%** |
| | EMMA | VLM | 71% | 88% | 94% | 85% | 83% | 67% | 82% |
| | EMAC+ | VLM | 79% | 88% | 93% | **90%** | 90% | 74% | 86% |
| No access to text env | MiniGPT-4 | VLM | 04% | 17% | 00% | 19% | 17% | 06% | 16% |
| | RL4VLM | VLM | 47% | 14% | 10% | 14% | 18% | 18% | 21% |
| | Idefics2 8B* | VLM | 75% | 77% | 74% | 69% | 67% | 50% | 69% |
| | VIPER$_{BC}$ | VLM+LLM | **80%** | 77% | 67% | 87% | **71%** | 53% | 72% |
| | VIPER$_{BC+RL}$ | VLM+LLM | **80%** | 77% | 77% | **92%** | **71%** | 53 % | 75% |
| | PRISM$_{BC}$ | VLM+LLM | **80%** | 77% | 81% | 87% | **71%** | 62% | 77% |
| | PRISM$_{BC+RL}$ | VLM+LLM | **80%** | **83%** | **84%** | **92%** | **71%** | **67%** | **80%** |

observations $o^t$ with textual descriptors $d_f^t$ obtained through PRISM's interactive goal-oriented perception process. The LoRA-enhanced policy $\pi_\theta$ is first optimized to mimic these expert demonstrations via a BC objective:

$$\mathcal{L}_{BC}(\theta) = -\mathbb{E}_{(d_f^t, \mathcal{H}, g, a^t) \sim \mathcal{D}_{\text{text}}} \left[ \log \pi_\theta(a^t | d_f^t, \mathcal{H}, g) \right]. \quad (6)$$

We then transition to online RL using Proximal Policy Optimization (PPO) (Schulman et al., 2017). Crucially, our dynamic QA-based perception remains active during the RL phase. Our LLM-based policy $\pi_\theta(a^t | d_f^t, \mathcal{H}, g)$ is optimized to minimize the PPO CLIP objective:

$$\mathcal{L}_{PPO}(\theta) = \hat{\mathbb{E}}^t \left[ \min(r^t(\theta)\hat{A}^t, \text{clip}(r^t(\theta), 1 - \epsilon, 1 + \epsilon)\hat{A}^t) \right] \quad (7)$$

where the advantage estimate $\hat{A}_t$ is derived from a learned value function $\hat{V}(d_f^t, \mathcal{H}, g)$, as in Carta et al. (2023).

## 4. Experiments

### 4.1. Experimental setup

**Environment:** Our experiments are conducted in two distinct environments: the ALFWorld simulated environment (Shridhar et al., 2021), which focuses on sequential interaction with household objects, and the Room-to-Room (R2R) environment (Anderson et al., 2018), dedicated to navigation within photo-realistic scenes. We perform all in-depth analyses and ablation studies in ALFWorld, while R2R is used to evaluate the versatility of our method across different visual and task domains. For each experiment, evaluation is performed on out-of-distribution test sets. Details regarding the environments are provided in Appendix B.1.

**Implementations:** We implement PRISM with Idefics2-8B (Laurençon et al., 2024) as the VLM for perception, as it offers the best trade-off between efficiency and performance

on POPE (Li et al., 2023). For both reasoning and sequential decision-making, we employ Mistral-7B (Jiang et al., 2023). While our main results (Section 4.2) focus on this configuration, we demonstrate the generalizability of our method to other LLM backbones in Section 4.3. Specifically, we use the frozen base LLM to leverage its inherent common-sense reasoning, while enabling fine-tuned LoRA adapters—trained following the method in Section 3.3—exclusively for decision-making. Details regarding the LoRA finetuning can be found in Appendix B.2.

**Baselines:** We compare PRISM with two families of approaches. The first family consists of agents trained with supervision from the ALFWorld textual environment (i.e., perfect description). This oracle supervision provides agents with perfect perception and access to privileged information not present in the visual observation, such as object IDs, states of occulted objects, and global environment dynamics. (More details are provided in Appendix B.5.1.) This includes LLM-based agents that operate solely on textual observations, such as ReAct (Yao et al., 2023), Reflexion (Shinn et al., 2023), DEPS (Wang et al., 2024), and Auto-Gen (Wu et al., 2023). It also includes VLM-based agents that process visual observations but are still supervised using these oracle textual descriptions, such as EMMA (Yang et al., 2024) and EMAC+ (Ao et al., 2025).

The second family comprises methods that do not rely on supervision from a textual environment or privileged information, learning instead directly from multimodal interactions. This family includes VLM-only agents, either zero-shot (MiniGPT-4 (Zhu et al., 2023)) or trained (Idefics2 8B (Laurençon et al., 2024) and RL4VLM (Zhai et al., 2024)). It also includes VLM+LLM agents, specifically VIPER$_{BC}$ and VIPER$_{BC+RL}$ (Aissi et al., 2025). Further implementation and training details are provided in Appendix B.2.

## 4.2. Main results on ALFWorld

We first study the evaluation results in ALFWorld for all approaches. As Table 1 shows, PRISM demonstrates notable superiority over methods without access to oracle textual descriptions. It achieves a considerable performance gain of approximately 60 pt compared to the MiniGPT-4 and RL4VLM baselines. PRISM $_{BC}$ and PRISM $_{BC+RL}$ also significantly outperform the Idefics2 8B model with respectively 9 pt and 12 pt gains on the average SR.

Notably, the improvement over VIPER highlights the importance of our DQA mechanism in bridging the gap between perception and action. The base configuration PRISM $_{BC}$ already surpasses VIPER$_{BC+RL}$ (77% vs 75%), effectively exceeding the performance of the latter even when it benefits from RL. The integration of RL into our approach, PRISM $_{BC+RL}$, further enhances these results by providing consistent gains across 4 out of 6 tasks compared to the BC-only version. This demonstrates the ability of our approach to successfully deal with RL in visual environments—a domain where baselines like RL4VLM often struggle. Specifically, RL fine-tuning leads to improvements in the "Look", "Clean", "Heat", and "Pick2" categories, resulting in an average SR of 80%. This progression enables PRISM to achieve performance levels comparable to EMMA (82%) and EMAC+ (86%), despite those models benefiting supervision from oracle textual environment.

Beyond these absolute scores, PRISM is distinguished by its operational consistency: while text-based methods such as ReAct and DEPS exhibit highly heterogeneous results, PRISM maintains strong homogeneity across different tasks.

## 4.3. Ablation studies

To analyze the contribution of each component within our framework, we conduct a series of ablation studies evaluating the **1) impact of PRISM architecture compared to a monolithic VLM**, the **2) impact of merging strategies for DQA**, and the **3) impact of generated questions and answers** on providing task-relevant support for the decision-making process.

### 4.3.1. IMPACT OF PRISM ARCHITECTURE

We compare PRISM to two baseline categories to assess the impact of decoupling perception from decision-making. First, we consider monolithic VLMs, namely Idefics2 8B (Laurençon et al., 2024) and Qwen3-VL 4B (Bai et al., 2025), where perception and decision-making are end-to-end. Second, we compare with static modular VLM-LLM architectures (i.e. (Aissi et al., 2025; Pan et al., 2024)), noted **Raw perception**, where the VLM naively describes the scene for the LLM without our interactive goal-oriented perception mechanism. To isolate the effects of modularity,

*Table 2.* **Impact of separating perception and decision-making on success rates.** PRISM outperforms monolithic baselines by 7–8% on average, with significant gains in complex tasks.

| | Pick | Look | Clean | Heat | Cool | Pick2 | Avg |
|---|---|---|---|---|---|---|---|
| Idefics2 8B | 75% | **77%** | 74% | 69% | 67% | 50% | 69% |
| Raw perception (Idefics-Mistral) | 80% | 67% | 74% | 82% | 62% | 53% | 70% |
| PRISM $_{BC}$ (Idefics-Mistral) | **80%** | **77%** | **81%** | **87%** | **71%** | **62%** | **77%** |
| Qwen3-VL 4B | 75% | 67% | 77% | 74% | 62% | 53% | 68% |
| Raw perception (Idefics-Qwen3) | 75% | **77%** | 71% | 74% | 62% | 53% | 69% |
| PRISM $_{BC}$ (Idefics-Qwen3) | **80%** | **77%** | **81%** | **82%** | **67%** | **62%** | **75%** |

we create comparable pairs based on the LLM used for decision-making: (1) PRISM-Mistral is compared to Idefics2 and Raw perception (Idefics-Mistral), and (2) PRISM-Qwen3 (with Qwen3 4B (Yang et al., 2025) as LLM) is compared to Qwen3-VL and Raw perception (Idefics-Qwen3). This ensures a fair comparison across models with the same number of trainable parameters.

**Results:** As shown in Table 2, PRISM achieves superior results compared to Raw perception and Monolithic VLM across both model variants, yielding an average improvement of 6 pt. This gain is particularly pronounced in complex tasks, with an increase of 9 to 12 pt on Pick 2 and an SR of 87% on Heat. A detailed comparison between Raw perception and Monolithic VLM reveals similar results between the two variants. This demonstrates that, while the decomposition of perception and reasoning is necessary, the performance gain does not stem from this structure, but from the enhancement of perception through the extraction of task-relevant information. Finally, the PRISM architecture remains consistent even when changing the LLM, maintaining an average SR of 77% and 75% across tasks.

For the remaining ablation studies, we use the Idefic-Qwen3 variant of PRISM, trained with BC.

### 4.3.2. IMPACT OF MERGING STRATEGY FOR DQA

To evaluate how PRISM integrates additional information into the initial scene description ($d_i^t$), we compare it with two alternative merging methods. The first, CONCAT, appends the generated questions and answers to the initial description, mirroring the naive approach used by Discuss-Nav (Long et al., 2023). The second, QA-Only, discards the initial description entirely and relies solely on the QA pairs for decision-making.

**Results:** As shown in Table 3, our method consistently outperforms both the QA-only and CONCAT approaches, achieving a performance gain of 3 to 5 pt over previous work. This improvement is stronger in high-complexity tasks such as Pick2, validating the robustness of our architecture.

*Table 3.* **Impact of Information Merging Strategies.** LLM-merge PRISM outperforms rigid concatenation and QA-only baselines by up to 5% on average, demonstrating the necessity of synthesizing global context with task-specific details.

| | Pick | Look | Clean | Heat | Cool | Pick2 | Avg |
|---|---|---|---|---|---|---|---|
| CONCAT | 80% | 77% | 74% | 82% | 67% | 50% | 72% |
| QA-Only | 80% | 77% | 77% | 74% | 62% | 53% | 70% |
| LLM-merge (PRISM) | 80% | 77% | 81% | 82% | 67% | 62% | 75% |

Our ablation confirms that while QA-derived information is crucial, it is insufficient on its own, as the QA-only variant plateaus at an average SR of 70%. Furthermore, the naive concatenation of the original descriptions and QA pairs (CONCAT) proves suboptimal. As shown in Appendix B.6.4, CONCAT generates significantly longer and more redundant inputs. This verbosity not only increases computational cost but also introduces noise that obscures task-critical information, complicating the decision-making process.

In contrast, PRISM performs a targeted synthesis of both sources, extracting and merging their most pertinent elements. This refined input structure—both concise and comprehensive—streamlines information processing and facilitates more effective reasoning by the model.

### 4.3.3. IMPACT OF GENERATED QUESTIONS & ANSWERS

We evaluate the impact of generating questions and answers by comparing it with two reference configurations: Oracle questions (Oracle Q), which reproduces the DiscussNav setting (Long et al., 2023) using predefined questions, and Oracle questions and Answers (Oracle QA), which provides ground-truth pairs. For these baselines, questions are drawn from DialFRED annotations (Gao et al., 2022) and answers are extracted from the textual environment. We assess our approach based on the SRs in six tasks, see Table 4.

**Results:** Based on Table 4, Oracle Q and Oracle QA configurations achieve the same results across all tasks, suggesting that the quality of the responses provided by the VLM are highly effective for task completion. However, a comparative analysis between Oracle Q and PRISM highlights stable performance on four tasks, contrasted by a slight drop for the Heat and Cool. This leads to a marginal average drop of 2pt on average, attributable to the varying relevance of the generated questions. Although the qualitative analysis of Oracle and PRISM questions in Appendix B.6.3 confirms that the generated questions are coherent with task goals, they rely on common sense rather than specific environment dynamics. As shown in Example 1 of Appendix B.6.4, the Oracle questions are hardcoded with environment priors, specifically asking for a microwave, whereas the LLM asks about general heating devices such as an oven or stove. This reliance on general knowledge instead of rigid, environment-specific heuristics explains the marginal 2pt performance gap, but it highlights that PRISM remains effective with-

out being tethered to the predefined priors of ALFWorld. Further analysis regarding the relevance of these generated questions can be found in Appendix B.6.3

*Table 4.* **Success rate comparison between** PRISM **and human-annotated oracle questions and answers across tasks**: PRISM achieves identical SRs on 4 out of 6 tasks with a marginal 2% overall difference compared to oracle questions and answers.

| | Pick | Look | Clean | Heat | Cool | Pick2 | Avg |
|---|---|---|---|---|---|---|---|
| Oracle Q | 80% | 77% | 81% | 87% | 71% | 62% | 77% |
| Oracle QA | 80% | 77% | 81% | 87% | 71% | 62% | 77% |
| PRISM | 80% | 77% | 81% | 82% | 67% | 62% | 75% |

## 4.4. Model analysis

To thoroughly evaluate interactive goal-oriented perception, we compare two configurations: **Raw perception** (see Section 4.3.1), and **Goal-aware perception**, where the goal is included within the VLM prompt to condition the description. We evaluate these models on task performance and the accuracy of their generated descriptions against ground-truth from the parallel textual environment. This quantitative analysis uses lexical metrics (RougeL (Lin, 2004), METEOR (Banerjee and Lavie, 2005)), semantic evaluation with BERTScore (Zhang et al., 2020) and an LLM-as-a-judge approach using the GPT-5 API (the evaluation prompt is available in Appendix A.1).

**Results: Task performance:** As shown in Table 5, the Interactive goal-oriented perception approach (PRISM) significantly leads with a 75% average SR, outperforming Raw perception (68%) and the Goal-aware perception baseline (64%). Interestingly, simply conditioning the VLM on the task goal without a structured QA loop degrades performance by 4%, likely because forcing goal alignment induces hallucinations or deviates from the VLM's standard descriptive training. In contrast, the interactive nature of PRISM effectively mitigates these errors, particularly in Clean (+10%) and Heat (+8%) tasks. These results underscore that targeted, interactive information acquisition is far more robust than simple goal-conditioning for complex sequential tasks.

**Results: accuracy of generated descriptions** As illustrated in Figure 3, our interactive goal-oriented perception approach (PRISM) significantly outperforms both baselines, achieving substantial gains over Raw Perception in RougeL (+0.082), METEOR (+0.052), and BERTScore (+0.068). Specifically, in the LLM-as-judge evaluation, our method achieves a 46.6% preference rate, representing a +20.7 pt lead over the Raw perception baseline (25.9%). These statistically significant results ($p < 0.005$) demonstrate that the iterative interaction effectively enforces description accuracy. In contrast, Goal-aware perception consistently degrades performance and triggers hallucinations; as shown in Appendix Figure 10, focusing strictly on the goal leads

*Table 5.* **Impact of the Interactive goal-oriented perception on success rate:** PRISM achieves the highest average performance (75%), consistently outperforming both baselines.

| | Pick | Look | Clean | Heat | Cool | Pick2 | Avg |
|---|---|---|---|---|---|---|---|
| Raw perception | 75% | **77%** | 71% | 74% | 62% | 53% | 69% |
| Goal-aware perception | 75% | 67% | 74% | 69% | 48% | 50% | 64% |
| PRISM (our) | **80%** | **77%** | **81%** | **82%** | 67% | **62%** | **75%** |

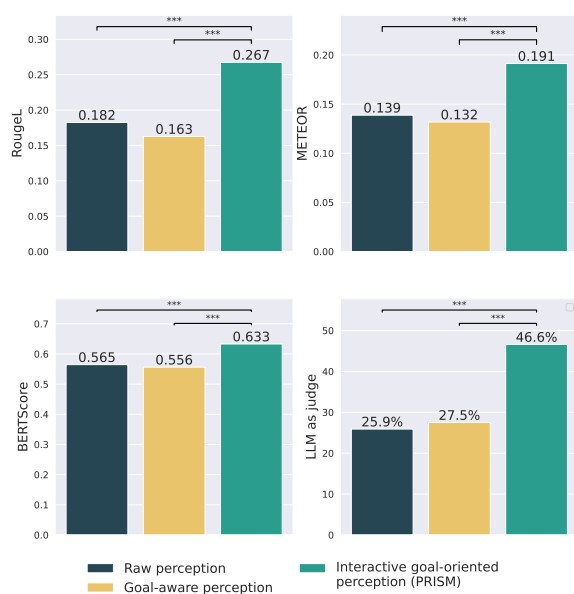

*Figure 3.* **Scene description quality compared to environment ground truth.** Our Interactive goal-oriented perception significantly outperforms other perception methods, demonstrating that the LLM-VLM Interactive goal-oriented perception enhances description accuracy. In contrast, Goal-aware perception reduces description quality. (*** indicates $p < 0.005$).

the VLM to describe non-existent objects. This highlights that goal-conditioning requires a dedicated grounding mechanism—like our QA interaction—to prevent the model from prioritizing perceived task relevance over environmental reality, which is detrimental to reliable downstream execution (as reflected in the lower task success rates in Table 5). Further qualitative comparisons and additional results across alternative VLM backbones are provided in Appendix B.6.1.

### 4.5. Extension to vision-and-language navigation tasks

**Baselines:** To evaluate the versatility of PRISM, we extend our analysis to the R2R navigation benchmark using BC on expert trajectories, comparing our architecture against LangNav (Pan et al., 2024), NavGPT (Zhou et al., 2023), and DiscussNav (Long et al., 2023). We report results from the original paper and our own reimplementations, denoted as LangNav* and DiscussNav*, which are built using the same model as our proposed architecture. For DiscussNav*, since

*Table 6.* **Performance on R2R.** Results marked with * are our reimplementation. PRISM achieves the best results in OSR, SR, and SPL compared to other methods, demonstrating superior navigation accuracy and efficiency.

| | NE↓ | OSR↑ | SR↑ | SPL↑ |
|---|---|---|---|---|
| Langnav | 7.10 | 45.00 | 34.00 | 29.00 |
| NavGPT | 6.46 | 42.00 | 34.00 | 29.00 |
| DiscussNAV | **5.32** | 61.00 | 43.00 | 40.00 |
| DiscussNAV* | 7.66 | 50.40 | 35.52 | 31.04 |
| Langnav* | 7.49 | 45.52 | 32.08 | 22.50 |
| PRISM $_{BC}$ (our) | 6.23 | **65.38** | **46.08** | **43.14** |

the original corpus of questions is unavailable and the full framework requires interaction with four distinct experts, we adapt the approach by using our own set of questions and concatenating the QA to the original scene description. With this adaptation, we maintain a consistent backbone and a controllable setting across all evaluations, ensuring a fair comparison to a standard prompt-augmentation approach, using identical visual information.

**Metrics:** We evaluate performance using standard navigation metrics: Navigation Error (NE), measuring the average distance to the goal at the stop position; Success Rate (SR), requiring the agent to stop within 3m of the goal; Oracle Success Rate (OSR), which counts a success if the agent passes within 3m of the goal at any point; and SR penalized by Path Length (SPL), which weights the success rate by the efficiency of the path.

**Results:** As shown in Table 6, PRISM achieves an SR of 46.08% and an OSR of 65.38%, and consistently surpasses established models like LangNav, NavGPT, and the original DiscussNAV —despite its access to multiple expert models and oracle questions—, as detailed in the Appendix B.5.2. This superior navigation accuracy is further reflected in an SPL of 43.14, the highest across all tested methods, indicating that PRISM navigates more successfully and with more efficient path execution. The most telling result emerges from the comparison with our DiscussNAV* reimplementation: despite both models using identical interactive questions, PRISM exceeds this baseline by over 10pt in SR and significantly reduces NE from 7.66 to 6.23. This gap reveals a fundamental architectural advantage; standard prompt concatenation introduces excessive noise, forcing models to process unrefined data during decision-making. By contrast, PRISM effectively structures sensory feedback, proving that the primary bottleneck in interactive navigation is not only the acquisition of information, but the ability to filter out noise and leverage task-relevant cues for superior spatial reasoning and goal reaching.

## 5. Conclusion

In this work, we introduced PRISM, a modular framework that addresses the perception-decision-making gap in embodied agents through a dynamic question–answer (DQA) pipeline. By tightly coupling a Vision-Language Model (VLM) for perception with a Large Language Model (LLM) for reasoning and decision-making, our approach enables the agent to dynamically identify and extract task-critical information that is often missed by standard goal-agnostic scene descriptions. Our evaluations on ALFWorld and Room-to-Room benchmarks show that guided perception outperforms monolithic VLMs and traditional systems where these modules remain isolated. These results suggest that interleaved reasoning effectively compensates for perceptual limitations, offering a robust path toward autonomous embodied agents. Future work will examine the challenge of integrating these interleaved perception-reasoning capabilities with lower-level control, such as robotics settings, to bridge the gap between decision-making and physical action.

## Impact Statement

This paper presents work aimed at advancing language-conditioned reasoning from visual observation through VLM-LLM architecture. While our approach improves the interpretability and autonomy of general-purpose robots, it also inherits the biases of large-scale pre-trained models. Beyond these broader considerations for autonomous systems, there are no immediate societal consequences of this research that must be specifically highlighted here.

## Acknowledgments

Experiments presented in this paper were carried out using the HPC resources of IDRIS under the allocations 2025-[AD011015093R1] and 2025-[AD011016980R1] made by GENCI. This work was supported by the European Commission's Horizon Europe Framework Programme under grant No 101070381 (PILLAR-robots), by RODEO Project (ANR-24-CE23-5886), by PEPR Sharp (ANR-23-PEIA-0008, ANR, FRANCE 2030) and by Cluster PostGenAI@Paris (ANR-23-IACL-0007, FRANCE 2030).

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

# A. Method details

This section provides technical details regarding our pipeline to ensure reproducibility. We describe the interactions between the perception and reasoning modules, as well as the communication protocols used during the Goal-Oriented Perception phase.

## A.1. Prompts

We employ structured templates to maintain consistency across different tasks and stages of the pipeline. Below, we detail the specific instructions used for each phase.

**Initial description:** prompt used by the VLM to generate the initial description of the observation $o^t$.

---

**Prompt: Initial Description**

Describe the scene in detail

---

**Question Generation:** Prompts used by the reasoning module ($R$) to identify missing information in $d_i^t$ relative to goal $g$.

---

**Prompt: Question Generation**

You are a robot and you need to achieve a task.
Your current observation is: $d_i^t$
Your goal is: $g$
Task:
- Generate a set of questions to obtain missing information and missing objects of the goal in the observation that is necessary to achieve the goal. If the information is present in the observation, don't ask question about it, ask only about missing informations in the observation.
- If no information is missing, respond with: "I have all the information."
- The question must: * Be related only to the given observation, and goal (not other parts of the house). * Begin with: "Do you see ..."
Output format:
A valid JSON dictionary like this and complete the questions:
{{"question1": "Do you see ... ?",...}}

---

**Visual Question Answering (VQA):** Instructions provided to the perception module ($VP$) to answer questions $\mathcal{Q}^t$ about missing or additional information (i.e., extracting precise visual attributes from the image).

---

**Prompt: Visual Question Answering**

Look at the image and answer the question with a detailed explanation. Do not answer with just yes/no: $q_i^t$

---

**Description Refinement:** The template used to aggregate the initial description $d_i^t$ with the questions $\mathcal{Q}^t$ and associated answers $\Lambda^t$ into a final description $d_f^t$.

---

**Prompt: Description Refinement**

You are an assistant that merges visual information into one coherent scene description.
Your task is to:
1. Take the initial description of the scene.
2. Integrate all new information provided in the question–answer pairs.
3. Remove contradictions and keep only information confirmed by the answers.
4. Produce ONE final, clean, coherent, and natural description of the scene.
5. Do NOT mention the questions, the Q/A process, or uncertainty.
6. Only describe what is explicitly confirmed.

---

---

Here is the initial description: $d_i^t$

Here are the question–answer pairs: $\mathcal{Q}^t, \Lambda^t$

Now rewrite the scene into a SINGLE enriched description that combines all confirmed details. Make it concise, factual, and fluent. Do not invent new objects or properties.

---

In addition to our main pipeline, we also provide the specific prompts used for the Goal-Aware Perception baseline from Section 4.4—where the goal is directly injected into the perception module—and the LLM-as-a-Judge template, used to evaluate how well the generated description aligns with the ground truth of the environment (from Section 4.4).

---

**Baseline: Goal-Aware Perception**

You are a robot and you need to accomplish the following goal: $g$.

Task: Identify the key elements to reach the goal. Then, describe the image.

---

**Evaluation prompt: LLM as judge**

Your task is to determine which of three candidate descriptions more close to the GT.

### Instructions

- Carefully compare each candidate with the GT.
- Focus on:
1. The presence of all objects mentioned in the GT.
- Ignore minor stylistic or grammatical differences.
- Choose **one** candidate that best matches the GT.
- If two or more descriptions are equally aligned with the GT, output "tie".
- Do not provide any reasoning or explanation.

### Instructions

1. Compare each detail in Text A, Text B and Text C against the GT.
2. Do not return empty output
3. Only output one of these options exactly:
- "Text A is more aligned"
- "Text B is more aligned"
- "Text C is more aligned"

Goal Text: GT

Text A: Raw perception

Text B: Goal Aware perception

Text C: Interactive goal-oriented perception

Best aligned description [Text A/Text B/Text C]:

---

# B. Experiments

## B.1. Environment

**Room-to-Room (R2R):** The R2R dataset is a benchmark for Vision-and-Language Navigation (VLN) based on the Matterport3D simulator. It uses real-world indoor panoramas where an agent must follow natural language instructions. The action space is discrete, consisting of: Turn Left, Turn Right, Move Forward, and Stop. Success is measured by the agent's ability to reach the target location within a specific distance threshold.

**ALFWorld:** ALFWorld is a synthetic environment combining AI2-THOR and TextWorld for multi-step household tasks. It requires agents to perform object manipulation (e.g., cleaning, heating) through a text-based action space. Success depends on the agent's ability to identify task-relevant objects and execute a sequence of interactions in the correct order.

*Table 7.* Episode counts for Train and Val Unseen splits.

| Environment | Train Episodes | Val Unseen Episodes |
|---|---|---|
| Room-to-Room (R2R) | 14039 | 2,349 |
| ALFWorld | 3,553 | 134 |

## B.2. Implementations details

**Training dataset**  For data collection in ALFWorld, we use the native rule-based bot provided by the environment. Given this bot's success rate of 70% over the training tasks, we ensure the quality of demonstrations by training our agents exclusively on successful trajectories. During data collection, we run PRISM interactive goal-oriented perception for each transition to turn the environment observations into textual descriptions. We then train the decision-making component with BC using this data (thus generating textual descriptions only once). Following this initial supervised phase, we perform RL across the entire training set, starting from the BC weights.

For the R2R environment, we use the bot provided by R2R. This bot uses the Dijkstra shortest-path algorithm to find the optimal path from the initial position to the goal. We use the same BC training scheme as in ALFWorld: we compute textual descriptions once over the entire training set and then train the decision-making component.

**Implementation Details**  We update the LLM used as sequential decision-making component by applying LoRA adapters exclusively to the attention layers ($W_q, W_v$). During BC training, we use a learning rate of $10^{-4}$. For online RL with PPO, we reduce the learning rate to $5 \times 10^{-6}$ to prevent catastrophic forgetting. More details about these hyperparameters can be found in Table 8. Following training, we keep the checkpoint with highest SR over training tasks as our final checkpoint used for evaluation.

*Table 8.* Hyperparameters for LoRA adaptation, BC pre-training, and PPO fine-tuning.

| Category | Parameter | Value |
|---|---|---|
| LoRA Configuration | Target Layers | $W_q, W_v$ |
| | Rank ($r$) | 32 |
| | Alpha ($\alpha$) | 16 |
| **Context** | History Transition Length ($|\mathcal{H}|$) | 5 |
| **BC pre-training** | Learning Rate | $10^{-4}$ |
| | Optimizer | AdamW |
| **RL fine-tuning** | Learning Rate | $5 \times 10^{-6}$ |
| | PPO Clip ($\epsilon$) | 0.1 |
| | Entropy Coefficient | 0.001 |
| | Gamma ($\gamma$) | 0.99 |

## B.3. RL fine-tuning efficiency

We track the training progress by monitoring the SR across different tasks. Figure 4 shows the evolution of SR during the RL fine-tuning on ALFWorld.

The training curves show that applying PPO consistently improves SR. However, the scale of this improvement varies significantly between tasks. These curves corroborate the results from Table 1, particularly regarding the Pick task, where only little improvement is observed. For this specific task, the SR stagnates, hovering around 0.80 for a significant portion of the training episodes before a late increase.

## B.4. Inference Latency

We report in Table 9 the per-step inference latency of PRISM and Raw Perception (No QA), alongside their SR and average number of steps executed per episode on ALFWorld. All measurements are collected on a single NVIDIA H100 GPU.

PRISM incurs a per-step latency approximately $3\times$ higher than Raw Perception, due to the three sequential forward passes involved at each step: question generation, visual question answering, and description refinement. This overhead is however offset at the episode level: PRISM executes 18% fewer environment steps on average (17.1 vs. 20.9), resulting in fewer redundant or erroneous actions. Combined with a 6-point gain in SR, these results show that the additional per-step cost is compensated by more reliable action selection across the full episode.

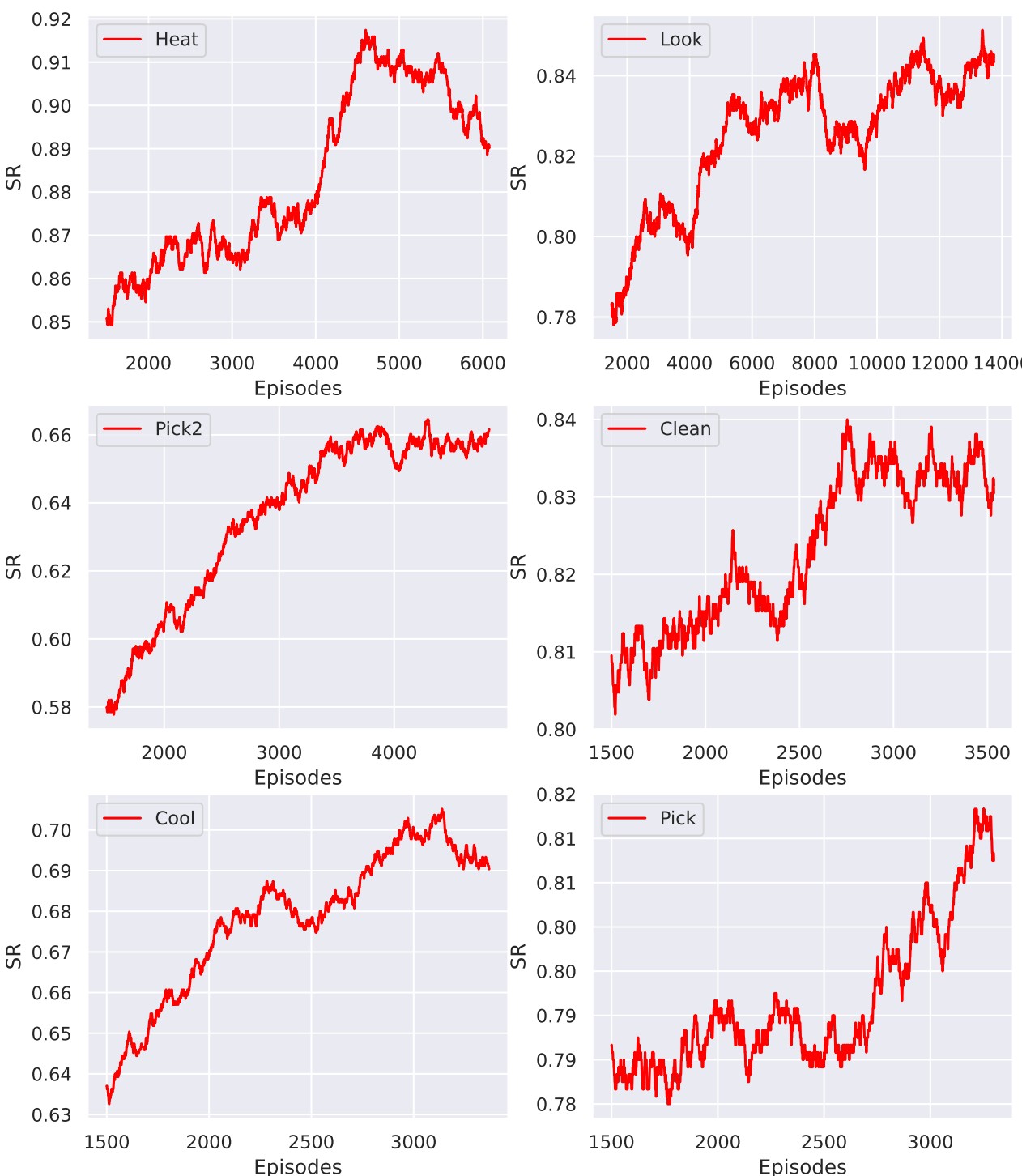

*Figure 4.* **Evolution of the success rate (SR) during RL fine-tuning across the six ALFWorld tasks.** The x-axis represents the number of training episodes, while the y-axis indicates the SR achieved.

*Table 9.* **Inference latency, SR and average steps on ALFWorld.**

| Metric | Raw Perception | PRISM (Ours) |
|---|---|---|
| Success Rate | 69% | 75% |
| Avg. Steps Executed | 20.9 | 17.1 |
| Inference Latency (s/step) | 2.14 | 6.35 |

## B.5. Baselines

### B.5.1. BASELINES ON ALFWORLD

This section details the ALFWorld baselines evaluated in our experiments (see Section 4) that utilize ground-truth supervision from the textual environment. This category includes methods such as ReAct (Yao et al., 2023), Reflexion (Shinn et al., 2023), DEPS (Wang et al., 2024), EMMA (Yang et al., 2024), and EMAC+ (Ao et al., 2025). Unlike approaches that operate on visual input, these agents are trained and evaluated with access to privileged state information provided by the simulator in textual form. This supervisory signal includes perfect symbolic metadata—such as the unique internal IDs of all objects, the presence and states of occluded items not in the agent's view, and explicit goal destination IDs—which is impossible to infer from pixel-level observations alone.

An example of this informational advantage is illustrated in Figure 5. The textual observation for the scene lists objects such as spraybottle 1, toiletpaper 1, plunger 1, and cloth 1. In the corresponding visual scene, only the spraybottle is directly visible. The toiletpaper is occluded behind it, and the plunger and cloth are not present in the image at all. Critically, the textual supervision not only reveals the existence and precise identifiers of these non-visible objects, but also provides their exact labels (e.g., 1). This information creates a perfect, lossless mapping to the simulator's discrete action space (e.g., ake plunger 1 from toilet 1). A purely visual agent cannot perceive these absent or hidden objects, nor can it infer their abstract IDs from pixels. This fundamental disparity in environmental access—where text-supervised agents receive privileged symbolic truth—explains the substantial performance gap between these baselines and PRISM, which must reason and act based solely on partial, egocentric visual observations.

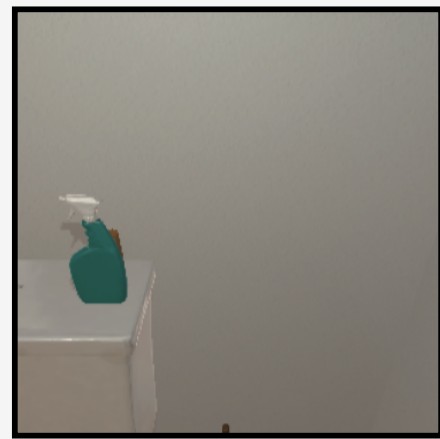

Textual environment description:
You arrive at loc 8. On the toilet 1, you see a spraybottle 1, a toiletpaper 1, a plunger 1, and a cloth 1.

Possible Actions:
'go to countertop 1', 'go to cabinet 1', 'go to cabinet 2', 'go to sink 1', 'go to cabinet 3', 'go to sinkbasin 1', 'go to handtowelholder 1', 'go to handtowelholder 2', 'take spraybottle 1 from toilet 1', 'take toiletpaper 1 from toilet 1', 'take plunger 1 from toilet 1', 'take cloth 1 from toilet 1', 'inventory', 'look', 'examine toilet 1'

*Figure 5.* An example of a textual observation returned by ALFWorld Textual Environment

### B.5.2. BASELINES ON R2R

In R2R, existing baselines primarily address environmental ambiguity through static perception pipelines. For instance, NavGPT employs a static perception module, using InstructBLIP as its VLM and GPT-3.5 via an API for decision-making. It thus relies on the powerful but generic reasoning of a large, zero-shot LLM. In a parallel architectural approach, LangNav also decouples perception from decision-making, but opts for a lightweight LLM specifically fine-tuned for navigation. Alternatively, DiscussNav queries a predefined corpus of oracle questions, creating a separation where perception (using a VLM and an object-detection model (Zhang et al., 2023)) gathers information, and a static set of questions retrieves answers. While distinct in implementation, these methods share a core limitation: their perception-and-information-gathering mechanisms are fixed and non-adaptive, operating independently from the navigation agent's immediate needs or

the evolving context.

## B.6. Additional results

### B.6.1. DESCRIPTION QUALITY

We extend our analysis of scene description quality by evaluating the PRISM framework with alternative VLM backbones, specifically employing InternVL for perception while maintaining Mistral as the reasoning module. This complements the main results where Idefics served as the primary VLM. As illustrated in Figure 6, our Interactive goal-oriented perception approach (PRISM) significantly outperforms both Raw and Goal-aware perception baselines even when using this alternative backbone, achieving a dominant 66.9% preference in the LLM-as-a-judge evaluation and superior scores across RougeL, METEOR, and BERTScore ($p < 0.005$). While the Goal-aware baseline using InternVL shows a marginal improvement in semantic alignment (BERTScore) over the raw output, it suffers from a decrease in description quality (RougeL), highlighting that a simple goal-directed prompt is insufficient. These findings confirm that the performance gains of PRISM are consistent across different VLM backbones.

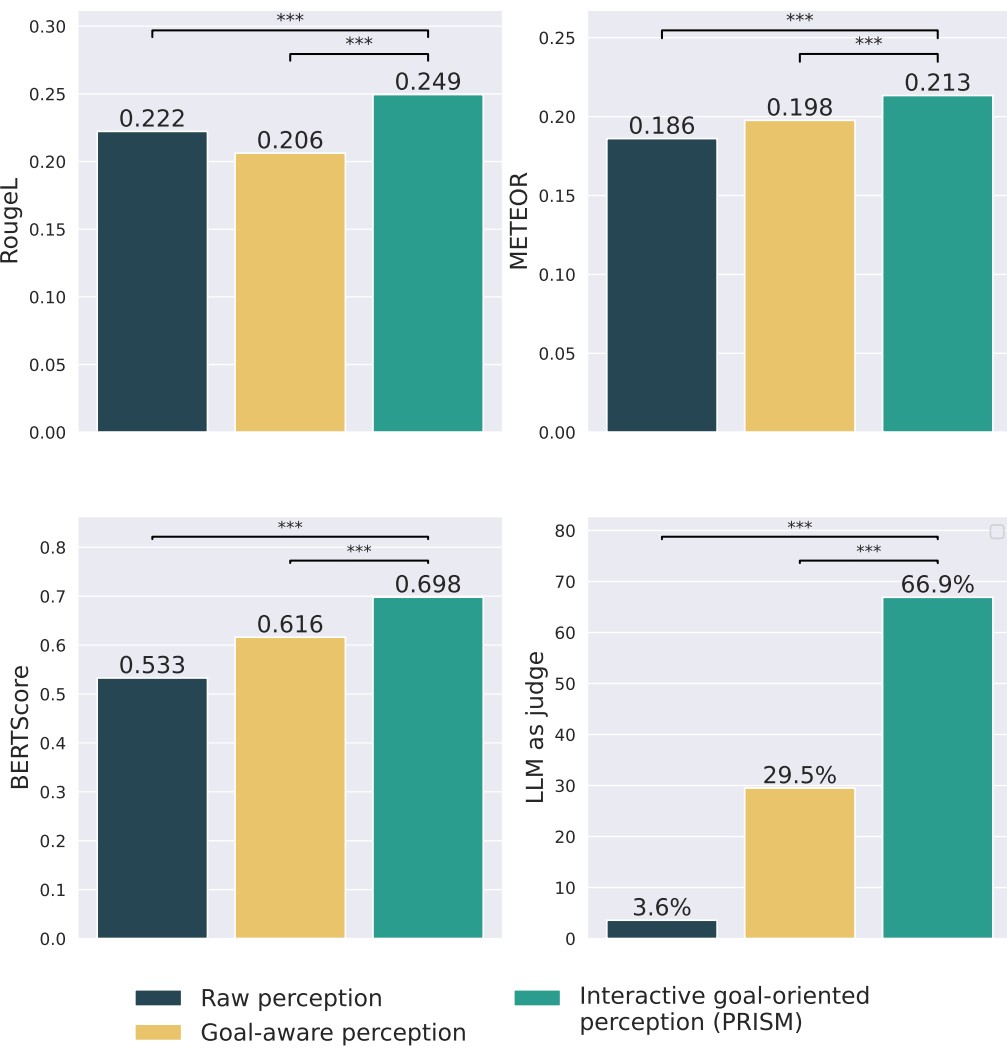

*Figure 6.* **Scene description quality compared to environment ground truth with InternVL.** Our Interactive goal-oriented perception approach significantly outperforms other perception methods. In contrast, Goal-aware perception reduces description quality. (*** indicates $p < 0.005$).

### B.6.2. VLM ANSWER ACCURACY

In this section, we evaluate the accuracy of Idefics2 responses to goal-directed questions over 1,000 transitions sampled from ALFWorld. Ground-truth answers are derived from the symbolic state of the textual environment. We report precision, recall, and F1-score per task category in Table 10.

*Table 10.* **VLM answer accuracy per task category on ALFWorld.** Metrics are computed by comparing Idefics2 responses against ground-truth answers from the textual environment over 1,000 transitions.

| Metric | Pick | Look | Clean | Heat | Cool | Pick2 | Avg |
|---|---|---|---|---|---|---|---|
| Precision | 0.85 | 0.91 | 0.96 | 0.96 | 0.83 | 0.95 | 0.91 |
| Recall | 0.89 | 0.90 | 0.95 | 1.00 | 0.87 | 0.97 | 0.93 |
| F1 | 0.87 | 0.90 | 0.95 | 0.98 | 0.85 | 0.96 | 0.92 |

Idefics2 achieves an average F1-score of 0.92. Performance is strongest on Heat and Clean, where task-relevant objects are spatially prominent, and slightly lower on Cool and Pick due to greater visual ambiguity and object occlusion. The average error rate of 8% does not translate into proportional degradation in task SR, consistent with the Oracle comparisons in Table 1. This robustness is attributable to the description refinement step, which filters inconsistent information before it reaches the decision-making module, and to the sequential structure of the tasks, which allows the agent to recover from isolated perceptual errors over subsequent steps.

### B.6.3. QUESTION GENERATION

**Impact of the number of questions:** In this section, we evaluate the impact of the number of questions used in our Interactive goal-oriented perception approach (using the Idefics-Qwen variant).

We first report in Figure 7 the mean number of questions asked by PRISM over the different environments and tasks. Results indicate PRISM asks an average of 2 to 3 questions.

We then evaluate PRISM when forcing the number of questions that must be generated. We do so by specifying the maximum number of questions that can be asked at each step. The results, detailed in Table 11, demonstrate that increasing the QA Budget from 1 to 3 leads to a consistent performance gain, with the average SR increasing from 70% to 74%. This trend suggests that the ability to ask multiple questions allows the LLM to better disambiguate task requirements, particularly in tasks such as Heat, where improvement is 8%.

Importantly, results show that PRISM still obtains the highest performance with dynamic question generation.

*Table 11.* **Impact of number of Questions on Performance**

| | Pick | Look | Clean | Heat | Cool | Pick2 | Avg |
|---|---|---|---|---|---|---|---|
| QA Budget = 1 | 75% | 77% | 74% | 74% | 62% | 58% | 70% |
| QA Budget = 3 | 80% | 77% | 77% | 82% | 67% | 62% | 74% |
| PRISM | 80% | 77% | 81% | 82% | 67% | 62% | 75% |

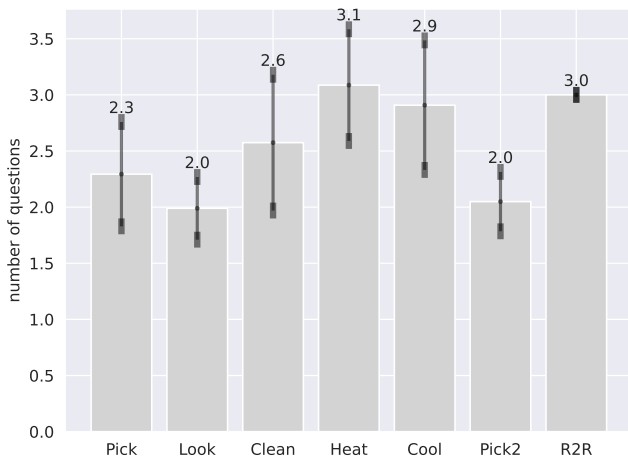

*Figure 7.* **Average number of questions asked by PRISM across different tasks and environments.**

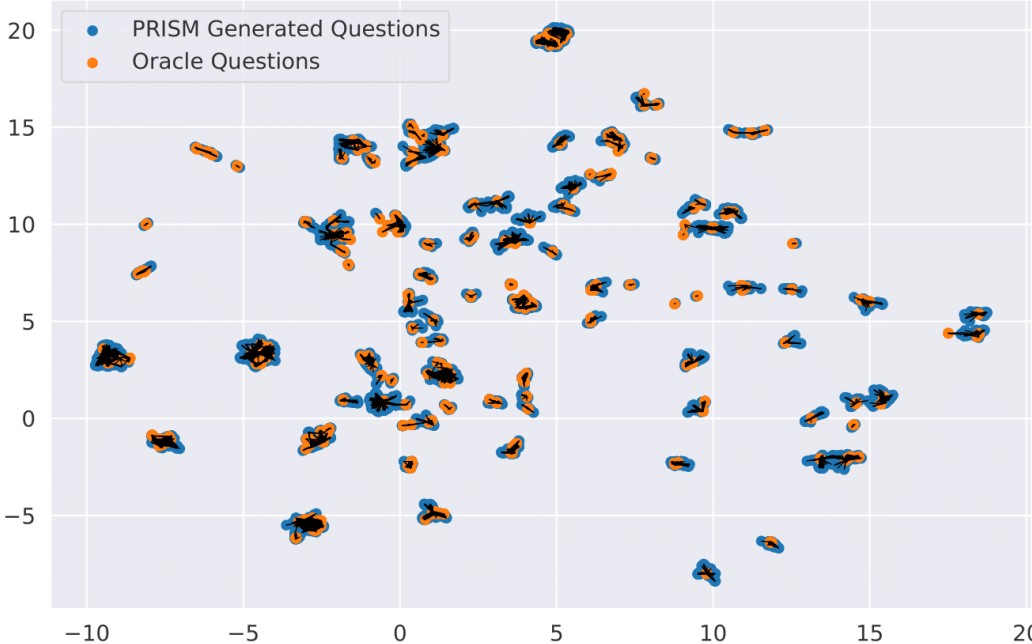

*Figure 8.* **Semantic alignment between Oracle and questions generated by PRISM.** T-SNE visualization of BERT embeddings along with lines between two questions associated with the same environment state. Results demonstrate that PRISM produces questions similar to the ones authored by humans.

**Relevance of generated questions:** To evaluate the quality of the generated questions, we perform a comparative analysis with the human-authored Oracle questions described in section 4.3.3. We first compute BERT embeddings for both the Oracle and questions generated by PRISM and visualize their distributions using t-SNE. As shown in Figure 8, the two sets are closely aligned within the semantic space, indicating a high degree of distributional similarity. Furthermore, by connecting points belonging to the same environment state with arrows, we observe that the generated questions remain close to their corresponding oracle reference. This consistent state-level alignment confirms that PRISM accurately identifies the same information needs as the human-authored questions.

### B.6.4. QUALITATIVE EXAMPLES

In this section, we provide qualitative results and examples of PRISM.

**Generated questions vs Oracle questions:** We observe in Example 1 that the generated questions are remarkably close to the Oracle ones, as they successfully identify the missing information in $d_i$ that is required to solve goal $g$. However, environment understanding remains limited. While the generated questions demonstrate some common-sense understanding from the LLM, the Oracle ones benefit from a clearer prior knowledge of the environment. For instance, in the heating task, the Oracle focuses on a "Microwave" and even queries its state (e.g., "is it empty?"), showing it already knows which specific heating device is available in that scene. In contrast, the generated questions remain broader, asking for a "stove or heating source" in general. This same distinction appears in cleaning tasks; where the Oracle might ask about a specific sink based on its prior mapping of the room, the LLM asks about "cleaning devices" generally. Ultimately, while the LLM correctly infers the necessary steps through logic and identifies what is missing from the initial description, the Oracle questions are more precise because they are tailored to the known affordances of that specific space.

---

**Example 1**

$d_i$**:** In this picture we can see a table, there are some plants, bottles, bowls and other things present on the table, in the background we can find a wall.
**Goal** $g$**:** Heat some egg and put it in the dining table
**Oracle questions:** [Do you see a egg in the image?, Do you see a Microwave in the image?, Do you see a dining table in the image?', 'is the Microwave empty?]
**Generated questions:** ['Do you see a stove or heating source?', 'Do you see an egg?', 'Do you see a plate or container to place the cooked egg on?', 'Do you see a dining table?']

---

**Example 2**

$d_i$**:** In this picture we can see a table, there is a toaster, bread, an apple and some other things present on the table, in the background we can find cupboards.
**Goal** $g$**:** Clean some tomato and put it in the microwave.
**Oracle questions:** [Do you see a plate in the image?, Do you see a dining table in the image?, Do you see a fridge in the image?', 'Is the fridge open?']
**Generated questions:** ['Do you see a plate that needs to be cooled?', 'Do you see a cooling source or method available, such as ice or cooler?', 'Do you see a dining table?']

---

**Example 3**

$d_i$**:** In this picture we can see a table, there are some bottles, bowls, spoons and other things present on the table, at the bottom there is floor.
**Goal** $g$**:** Cool some plate and put it in the dining table.
**Oracle questions:** [Do you see a egg in the image?, Do you see a Microwave in the image?, Do you see a dining table in the image?', 'Is the Microwave empty?]
**Generated questions:** ['Do you see a stove or heating source?', 'Do you see an egg?', 'Do you see a plate or container to place the cooked egg on?', 'Do you see a dining table?']

---

**Merging strategies:** In this analysis, we compare the merging strategy used by PRISM and the CONCAT strategy (see Section 4.3.2), notably used by DiscussNav. At an equivalent operational state, the CONCAT approach exhibits a significantly higher token volume, often doubling the prompt length; this makes processing the textual observation harder, as the model must disentangle the relationship between raw QA pairs and initial descriptions. In contrast, PRISM employs a merging mechanism that synthesizes these elements into a more compact and accurate representation. By eliminating semantic redundancy and maintaining a lower token count without sacrificing critical information.

---

PRISM **prompt in R2R**

You are an agent and you need to accomplish a task Your task is to: Walk down one flight of stairs and stop on the landing.
**Current Observation:**
Front: The scene consists of a pillar, floor mat, various objects on it, doors, windows, a chandelier, walls, and a railing on the left side. Additionally, there's a staircase visible, with landings between each set of steps, all stairs either lead to other floors or upward, and they have a handrail on both sides. At the top right corner of the image, there is a ceiling with lights.
Left: The scene consists of pillars, carpeted floor, various objects, and a fire extinguisher located on the left side. Above, lights illuminate the space, but there is no sign of a staircase or a handrail within view.
Right: The scene features a pillar, carpet on the floor, various objects, and a set of stairs with a handrail on one side, descending from the area where these items are located, under a lit ceiling.
Next action of the agent:

(**Number of token= 967**)

---

**CONCAT prompt in R2R**

You are an agent and you need to accomplish a task. Your task is to: Walk down one flight of stairs and stop on the landing.
**Current Observation:**
Front: In this picture, we can see a pillar, floor mat and some objects on it. We can see doors, windows, chandelier and wall. On the left side of the picture we have railing. At the top right corner of the image, we can also see ceiling with lights.
QA: [('Do you see any staircase leading downwards from where I am currently located?', ' yes, there is a visible staircase that leads downward from my current location.'), ('Do you see a handrail along the staircase that leads downwards?', ' Yes, there is a hand railing on both sides of the stairs.'), ('Is there a landing area visible between two consecutive steps while looking at the downward staircase?', ' Yes, there is a landing in-between the stairs as we can see from this picture.')]
Left: In this picture, we can see pillars, carpet on the floor and some objects. On the left side of the image there is a fire extinguisher. At the top of the ceiling there are lights.
QA: [('Do you see any staircase leading down from where you currently are?', ' No, I do not see any stairs in this room.'), ('Do you see a handrail or anything similar along the staircase?', ' No, there is no handrails on the stairs.')]
Right: In this picture, we can see a pillar, carpet on the floor and some objects. On top of the image there is ceiling with lights.
QA: [('Do you see any staircase leading down from the location where the pillar, carpet, and objects are?', ' Yes.'), ('Do you see a handrail or anything similar along the potential staircase?', ' Yes, there is a hand railing on one side of the stairs.')]
Next action of the agent:
(**Number of token=1764**)

---

**Interactive goal-oriented perception**   Finally, we provide a qualitative analysis of the whole Interactive goal-oriented perception in Figure 9 and Figure 10.

PRISM demonstrates a sophisticated ability to generate relevant, goal-oriented questions, particularly within the R2R dataset. As seen in Examples (1) and (2) of Figure 9, the model proactively inquires about the presence of a staircase and potential obstacles, which are essential pieces of information to complete a navigation task safely.

Examining ALFWorld reveals the limitations of Goal-aware perception. In several instances, this approach generates descriptions that are nearly identical to Raw perception, effectively ignoring the goal. Furthermore, this perception approach is prone to hallucinations when under pressure to find goal-related items; for instance, in Example (4) of Figure 10, the Goal-aware description falsely claims that there is "cd on the table," while PRISM's interactive perception later correctly identifies that no CDs are actually visible in the scene.

**Room 2 Room**

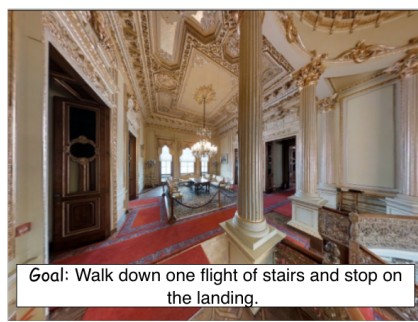

Goal: Walk down one flight of stairs and stop on the landing.

*Raw perception*

**Description:** In this picture we can see a pillar, floor mat and some objects on it. We can see doors, windows, chandelier and wall. On the left side of the picture we have railing. At the top right corner of the image we can also see ceiling with lights.

*Interactive goal-oriented perception*

**Question 1:** Do you see any staircase leading downwards from where I am currently located? No, there is no visible staircase that leads downward from my current location.
**Question 2:** Is there a landing area visible? No, there is no landing area as we can see from this picture.

**Description:** The scene consists of a pillar, floor mat, various objects on it, doors, windows, a chandelier, walls, and a railing on the left side. There's no staircase or landing area visible in the image.

(1)

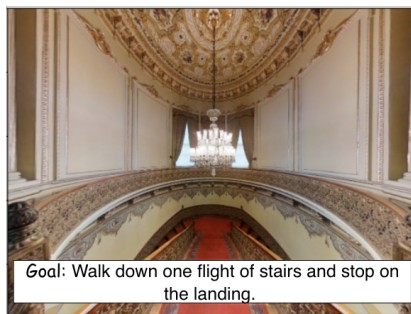

Goal: Walk down one flight of stairs and stop on the landing.

*Raw perception*

**Description:** In this picture we can see a chandelier, windows and curtains. At the bottom of the image we can find stairs with red carpet on it. On top of the roof there is some design.

*Interactive goal-oriented perception*

**Question 1:** Do you see any obstacles or irregularities on the stairs that could make walking difficult? Yes, there is an open paths for you to walk on the stairs without any obstacle.

**Description:** In the depicted scene, a grand staircase with a continuous red carpet runs from the lower part of the image towards the upper portion. The stairs have no obstructions making them safe to navigate. At the top of the stairs, a chandelier hangs from the ceiling, casting a warm glow over the area below.

(2)

Goal: Leave the room and turn left. Go inside the first door on the left and stand in front of the rug to the right of the tables.

*Raw perception*

**Description:** In this picture we can see a door, wall and glass windows. At the top of the image we can find lights on roof. On the left side of the window there is an object.

*Interactive goal-oriented perception*

**Question 1:** Do you see any obstacles blocking my path from reaching the door I mentioned earlier? No, there are no obstacles in your way to reach that door.
**Question 2:** Do you see any specific type of floor covering between me and the rug I am supposed to stand in front of? No, there is no rug or carpet on the ground here.

**Description:** The scene features a room with a wooden door equipped with a doorknob, allowing for easy access. There's a clear path leading to the door without any obstructions. And there is no visible rug in the scene.

(3)

*Figure 9.* **Qualitative examples of** PRISM **generated descriptions on R2R.**

**Alfworld**

*Figure 10.* **Qualitative examples of** PRISM **generated descriptions on ALFWorld.**

