# OpenReview forum: "PRISM: Perception Reasoning Interleaved for Sequential Decision Making."
_ICML.cc/2026/Conference — ICML 2026 regular_

### Official Review · Reviewer_Xcfr · 2026-03-11

**Soundness:** 3
**Presentation:** 4
**Significance:** 3
**Originality:** 3
**Overall Recommendation:** 4
**Confidence:** 4

**Summary:**

This paper introduces the PRISM framework, which aims to address the disconnection between perception (VLM) and decision-making (LLM) in embodied agents. By introducing a Dynamic Question-Answering (DQA) pipeline, the mechanism works as follows: the VLM provides an initial scene description, which is then "critiqued" by the LLM (the core of decision-making) based on the target task. The LLM proactively generates goal-oriented questions to query the VLM for missing information. Finally, the LLM synthesizes all question-answer interactions into a compact and precise textual description for decision-making. PRISM is evaluated on two benchmarks, ALFWorld and Room-to-Room (R2R), with results showing that PRISM significantly outperforms existing image-based models and decoupled VLM+LLM methods. The main contributions of the paper are the fully automatic DQA mechanism, an effective training strategy (BC+RL), a substantial and convincing set of ablation studies, and demonstrated empirical advantages across two benchmarks.

**Compliance With Llm Reviewing Policy:**

Affirmed.

**Final Justification:**

The author's rebuttal has resolved my concerns about inference efficiency and the errors in VLM perception. I will maintain my rating at 4 (Weak Accept).

**Key Questions For Authors:**

1.  **Regarding the Dynamic Depth of Interaction:** According to Algorithm 1, DQA appears to be a single-round interaction per timestep (generating all questions at once). How do the authors ensure this is sufficient for complex scenarios that may require step-by-step reasoning? Are there plans to introduce a dynamic, multi-round mechanism in future work?
2.  **Regarding Computational Overhead and Latency:** DQA is a multi-step pipeline (description -> questioning -> answering -> summarization). What is the approximate end-to-end inference latency per step on average in the ALFWorld or R2R environments? How does this overhead compare to baseline models like VIPER or Raw perception? While the paper discusses token redundancy, could the authors provide a comparison of the **absolute token overhead** between PRISM and baselines (e.g., average input+output tokens per step)? Have the authors considered the real-time challenges of deploying such a system on a physical robot?
3.  **Regarding Hallucinations During Interaction:** The paper nicely demonstrates that direct goal injection causes hallucinations. However, during the DQA interaction: ① Could the VLM also hallucinate when answering the LLM's fine-grained questions? ② Could the LLM introduce new, unsupported content during the summarization phase (summary hallucinations)? ③ Is there a risk of error accumulation or hallucination amplification throughout the interaction chain? Does the authors have any mechanisms to detect or mitigate these potential hallucinations?
4.  **Failure Case Analysis:** In Table 4, PRISM underperforms slightly on the "Heat" and "Cool" tasks compared to the Oracle. Could the authors provide a breakdown analysis of these failure cases? Are they primarily caused by visual hallucinations from the VLM, or by the LLM generating questions that are not specific enough?

**Limitations:**

See above.

**Strengths And Weaknesses:**

*   **Clearly Defined Problem:** The paper accurately identifies the "perception-decision gap" in VLM+LLM decoupled architectures for embodied AI.Specifically, the perception module fails to provide task-relevant, fine-grained and accurate information to the decision-making module. This problem is practical and challenging.
*   **Technical Innovation:** DQA of the PRISM framework is an good design. Unlike prior works that rely on predefined questions or oracle access (e.g., DiscussNav), PRISM's fully automatic question generation allows the agent to adaptively seek task-critical information. The synthesis step, which combines the initial description with QA pairs into a compact prompt, is also a well-validated design choice that helps reduce noise.
*   **Rigorous and In-depth Experimental Design:**
    *   The paper compares a wide range of baselines, including text-supervised (Oracle), pure VLM, and VLM+LLM models, which clearly positions the source of PRISM's advantages.
    *   The authors conduct comprehensive ablation studies. The work in Sections 4.3 and 4.4 is particularly solid, validating the decoupled architecture itself, the DQA information fusion strategy, the quality of question-answer generation, and the necessity of interactive perception. The comparison against "Goal-aware perception" strongly proves that simply injecting the goal leads to hallucinations, while DQA mechanism is key to solving this problem.
    *   The paper also analyzes the quality of the generated descriptions and the semantic alignment of generated questions with human-annotated ones. These analyses significantly enhance the persuasiveness of the conclusions.
*   **Significance of Contributions:** The core contribution (the DQA mechanism)is clearly articulated and validated. Furthermore, the framework's fully automatic nature (eliminating the need for manually predefined questions and answers) is an important aspect of its practical value.

**Weaknesses:**

*   **Computational Efficiency and Latency Not Discussed:** PRISM introduces a multi-step VLM-LLM interaction loop (VLM description -> LLM questioning -> VLM answering -> LLM summarization) at every time step. This introduces significant computational overhead and latency compared to traditional single-pass VLM descriptions or simple VLM+LLM concatenation. Although the paper discusses the token redundancy of the CONCAT strategy in Appendix B.5.3 and claims PRISM's summarization strategy reduces token count, it **lacks a comparison of the absolute token overhead between PRISM and baseline models (e.g., VIPER, Raw perception), and even more critically, does not quantify the time/latency overhead.** In real-world robotics or scenarios requiring rapid response, this is a practical flaw that cannot be ignored.
*   **Insufficient Discussion of Hallucinations During Interaction, and No Mechanisms to Detect or Mitigate Them:**
    *   **Existing Discussion:** The paper effectively demonstrates in Section 4.4 and Appendix Figure 10 that directly injecting the goal into the VLM (Goal-aware perception) causes hallucinations.
    *   **Missing Discussion:** The paper does not systematically analyze potential hallucinations during the DQA interaction process, including:
        ① Whether the VLM might hallucinate when answering fine-grained questions from the LLM (e.g., "Do you see X behind Y?").
        ② Whether the LLM, during the summarization phase, might introduce new content not present in any source information (i.e., summary hallucinations) when integrating the initial description and QA pairs.
        ③ Whether the entire interaction chain carries a risk of error accumulation or hallucination amplification.
*   **Uneven RL Gains:** The benefits from the Reinforcement Learning (RL) phase are uneven across different tasks. Furthermore, the model mainly relies on the zero-shot visual question-answering capabilities of a frozen VLM (e.g., Idefics2-8B), lacking end-to-end fine-tuning of the vision encoder. This could potentially limit its performance ceiling in complex environments requiring fine-grained physical interaction.

---

> ### Author Rebuttal · Authors · 2026-03-27
>
> # **Reviewer Response**
>
> We sincerely thank the reviewer for the exceptionally detailed and positive feedback. We are gratified that the reviewer recognized PRISM’s ability to bridge the **"perception-decision gap"** and valued our **"technical innovation"** in moving beyond predefined questions. We also appreciate the recognition of our **"rigorous and in-depth experimental design,"** specifically our analyses in **Sections 4.3** and **4.4** regarding goal-aware hallucinations. Below, we address the specific concerns raised in “weaknesses” (W1 to W4) regarding the Dynamic Interaction, latency, hallucinations, and failure case.
> ## **W1: Regarding the Dynamic Depth of Interaction**
> We thank the reviewer for the question regarding the interaction depth. In our current pipeline, we implement a single-round mechanism where the model generates all necessary questions at once. Our primary purpose is to demonstrate that even a single round of DQA significantly improves description quality and effectively reduces the perception-reasoning gap. While we agree that a dynamic, multi-round approach could provide further benefits, we have left the exploration of multi-round interactions for future work.
>
> ## **W2. Inference Latency and Operational Efficiency:**
> To address the concern regarding computational cost, we provide a systematic analysis comparing PRISM (using Idefics2 / Qwen3 4B) against the Raw Perception (No QA) baseline. All experiments were conducted on an NVIDIA H100 GPU.
> We emphasize that **PRISM operates at the high-level decision-making phase.** At this stage, the agent identifies the correct high-level action required to progress toward the goal. In such a context, **decision latency is significantly less critical than the accuracy and efficiency of the physical execution** that follows.
> **Results:**
> | Metric | Raw Perception (No QA) | PRISM (Ours) |
> | :--- | :---: | :---: |
> | **Success Rate (SR)** | 69% | 75% |
> | **Avg. Steps Executed** | 20.9 | 17.1 |
> | **Inference Latency ($s/step$)** | 2.14s | 6.35s |
>
> **Analysis of Decision Quality vs. Execution Speed**
> The data illustrates that while PRISM requires about 3 times more time per individual step, corresponding to 3 forward passes (to determine the questions, to determine the answers, and to generate the updated description), it significantly optimizes the overall task execution:
> * Step Efficiency: **PRISM executes 18% fewer high-level actions in the environment (17.1 vs. 20.9).** This indicates that the interactive QA process allows the agent to select more direct and effective actions.
> * Reduction of Irrelevant Decisions: **The additional "perception refinement time" (6.35s vs. 2.14s) prevents the agent from making irrelevant or redundant moves.** By reducing VLM error through targeted questioning, we ensure each high-level action is purposeful.
> * Performance Gain: **Prioritizing the quality of each choice results in a 6 pt absolute gain in Success Rate.** In complex tasks, **taking fewer, higher-quality actions is far more sustainable and successful** than taking many rapid, inaccurate ones.
> ## **W3. Regarding Hallucinations During Interaction**
> We thank the reviewer for the insightful question about hallucinations. In our pipeline, we specifically prompt the LLM to format its questions using a "Do you see...?" structure to reduce hallucinations in the VLM's answers. While measuring hallucinations is an open problem in the field, we show in **Figure 3**, through different NLP metrics, that our generated descriptions are significantly closer to the **Ground Truth (GT)** descriptions. This demonstrates that our design provides a more reliable scene description and effectively mitigates the risks of error accumulation or hallucination amplification throughout the interaction chain.
>
> ## **W4 Failure Case Analysis**
> We thank the reviewer for this question regarding the "Heat" and "Cool" tasks. We have conducted a failure analysis with examples in **Appendix B.5.3 (Example 1 and Example 3)**. Our analysis reveals that the slight performance gap compared to the Oracle is primarily due to environment priors. While the Oracle specifically asks for a **"microwave"** for heating or a **"fridge"** for cooling based on **ALFWORLD-specific priors**, **PRISM** generates more generalized queries. For instance, PRISM may ask about various heating devices (e.g., oven, microwave) or cooling objects (e.g., ice, fridge, cold water). Because PRISM does not rely on these specific environment biases, it occasionally results in a minor difference in efficiency, though it remains **more adaptable to general scenarios.**
>
> # **Conclusion:**
> We thank the reviewer again for these valuable suggestions. These clarifications regarding hallucinations, failure case, inference latency, and the depth of interaction have significantly strengthened the paper and will be included in the final version.

---

> > ### Author Rebuttal · Reviewer_Xcfr · 2026-04-04
> >
> > The author's rebuttal has resolved my concerns about inference efficiency and the errors in VLM perception.

---

> > > ### Author Response · Authors · 2026-04-07
> > >
> > > Thank you for your feedback. We are glad to hear that our responses addressed your concerns. Are there any remaining points that would prevent you from increasing your score?

---

### Official Review · Reviewer_Lz6B · 2026-03-13

**Soundness:** 3
**Presentation:** 3
**Significance:** 2
**Originality:** 2
**Overall Recommendation:** 4
**Confidence:** 4

**Summary:**

This paper introduces PRISM, a modular framework for embodied sequential decision-making that tightly couples a Vision-Language Model (VLM) for perception with a Large Language Model (LLM) for reasoning and action selection via a Dynamic Question-Answering (DQA) mechanism. At each time step, the VLM generates an initial goal-agnostic scene description; the LLM then identifies informational gaps relative to the task goal and generates targeted questions, which the VLM answers; finally, the LLM synthesizes the initial description and QA pairs into a compact, task-driven scene representation used for action selection. The LLM is fine-tuned with LoRA adapters via behavioral cloning followed by PPO-based reinforcement learning. The authors evaluate on ALFWorld (household manipulation) and Room-to-Room (vision-and-language navigation), showing improvements over prior VLM-only and decoupled VLM+LLM baselines that lack access to oracle textual environments. Ablation studies examine the architecture, merging strategies, question/answer quality, and description accuracy.

**Compliance With Llm Reviewing Policy:**

Affirmed.

**Final Justification:**

The authors’ rebuttal has addressed my concerns. However, the full experimental results are still missing, so I will maintain my positive score.

**Key Questions For Authors:**

1. How many additional VLM/LLM forward passes does DQA require per time step on average, and what is the wall-clock time overhead compared to the raw perception baseline?

2. Given the small evaluation set (134 episodes in ALFWorld), can you provide confidence intervals or results across multiple training seeds? For example, the 5-point improvement of PRISM BC+RL over VIPER BC+RL (80% vs. 75%) corresponds to roughly 7 additional successful episodes—how robust is this to random variation?

3. How often does the VLM provide incorrect answers to the generated questions, and how does error propagation through the synthesis step affect downstream task performance? A systematic error analysis (beyond the oracle QA comparison in Table 4) would substantially strengthen the paper.

4.  As VLMs improve (e.g., GPT-4V, Gemini), the initial descriptions may already capture most task-relevant information. Have you investigated whether DQA's marginal benefit diminishes with stronger VLM backbones? This would help clarify the longevity of the contribution.

**Limitations:**

The authors briefly discuss limitations in the Impact Statement, mentioning inherited biases from pre-trained models. However, several important limitations are not discussed: the computational cost of the multi-pass DQA pipeline, scalability to more complex real-world environments, the reliance on VLM answer quality (and what happens when it degrades), and the lack of analysis on how the approach handles adversarial or highly ambiguous visual scenes. The paper would benefit from a dedicated limitations section.

**Strengths And Weaknesses:**

Strengths：

The authors examine the concept of interleaving perception and reasoning through a closed-loop DQA mechanism, well-motivated by the observation that VLMs produce task-agnostic descriptions missing goal-critical details. The paper is clearly written with a helpful architectural diagram (Figure 2) and thorough ablation studies covering merging strategies (Table 3), oracle vs. generated questions (Table 4), and the hallucination-inducing effect of naive goal-conditioning (Table 5). The framework is fully automatic, generalizes across two domains (ALFWorld, R2R) and two LLM backbones, and the appendix provides sufficient detail for reproducibility.

Weaknesses:

1.  Limited originality. The authors claim to assess the central problem of bridging perception and decision-making, yet the solution chains four standard prompting calls (captioning → question generation → VQA → summarization) with no methodological innovation beyond prompt design. The delta over prior interactive QA work (DiscussNav, DialFRED) — replacing fixed questions with LLM-generated ones and concatenation with LLM summarization — is a natural, predictable extension.

2. Insufficient statistical rigor. The ALFWorld evaluation set has only 134 episodes (Table 7), so per-task gains of a few percentage points correspond to single-digit episode differences. No confidence intervals, standard deviations, or multi-seed results are reported, making it impossible to assess whether the claimed improvements are within noise.

3. Missing computational cost analysis. DQA requires 5–7 sequential model inferences per time step (vs. one for raw perception), yet no wall-clock times, token counts, or latency-accuracy trade-offs are discussed. This is a critical omission for an embodied agent framework.

---

> ### Author Rebuttal · Authors · 2026-03-27
>
> # **Reviewer Response**
> We thank the reviewer for their positive feedback. We are glad the closed-loop DQA mechanism was found **well-motivated** and the paper **clearly written.**  We also appreciate the recognition of our architectural diagram **(Fig. 2)** and ablation studies **(Tabs. 3–5)**. Below, we address the specific concerns raised in **Weaknesses (W1–W4)** regarding latency, statistical robustness, perception robustness, and scaling.
> ## **W1. Inference Latency and Operational Efficiency:**
> The concern regarding the computational overhead and latency compared to the raw perception baseline was also raised by Reviewer **bZL8**.
> To provide a focused and data-driven response, we have included a quantitative analysis of inference time ($s/step$) and trajectory efficiency in our response to Reviewer **bZL8 (W1)**. We kindly invite the reviewer to refer to that section.
> ## **W2. Statistical Robustness and Evaluation Set**
> We appreciate the reviewer’s concern regarding the evaluation scale. Our choice of environments was deliberate: **ALFWorld** serves as a controlled setting for detailed ablation and qualitative analysis, while **R2R** (with over 2,000 test episodes) allows us to demonstrate the statistical robustness of PRISM on a large scale.
> Please note that a single RL training run requires **approximately 100 hours per seed**, making new runs computationally infeasible within the short rebuttal window. However, to fulfill the reviewer's requests regarding the performance gains of PRISM, we performed additional statistical validation of our existing results using **McNemar’s tests** on the episode-level results for both R2R and ALFWorld.
> #### **A) Large-Scale Evaluation (R2R — 2,000 Episodes)**
> On the large-scale R2R dataset, we compared PRISM against two state-of-the-art baselines. The extremely low p-values confirm that our task-driven perception provides a highly consistent advantage across a vast population of episodes.
> | Comparison | **p-value** | **Statistical Significance** |
> |---|---|---|
> | **LanNav** vs. **PRISM** | $2.78 \times 10^{-7}$ | **Very significant** ($p < 10^{-6}$) |
> | **DiscussNAV*** vs. **PRISM** | $0.0013$ | **Significant** ($p < 0.01$) |
> #### **B) Episode-Level Significance (ALFWorld)**
> We further performed the McNemar test on ALFWorld to evaluate the significance of PRISM’s improvements over the **Raw Perception (No QA)** baseline.
> | Comparison | **p-value** | **Statistical Significance** |
> |---|---|---|
> | **Raw Perception** vs. **PRISM BC** | $0.0430$ | **Significant** ($p < 0.05$) |
> | **Raw Perception** vs. **PRISM BC+PPO** | $0.0215$ | **Significant** ($p < 0.05$) |
> **Analysis of Robustness:**
> * **Proven Significance:** For both our Behavioral Cloning (BC) and RL-enhanced (BC+PPO) versions, the p-values are below the standard threshold ($0.05$). This indicates that the success of PRISM represents a statistically likely architectural advantage.
> * **Consistency Across Scales:** By achieving significant p-values on both the large-scale R2R dataset ($p < 0.002$) and the fine-grained ALFWorld dataset ($p < 0.05$), we demonstrate that the DQA mechanism provides a robust and reproducible performance boost.
> ---
> ## **W3. Robustness to VLM Perception Errors**
> The reviewer raises an important point regarding the frequency of incorrect VLM answers and how they affect downstream performance. This was also a key point of discussion for Reviewer  **bZL8**.
> We have addressed this by performing error analysis (Precision, Recall, and F1-score) across 1,000 transitions to quantify hallucination rates and perception reliability. We kindly ask the reviewer to refer to our response to **Reviewer bZL8 (W2)**.
> ## **W4. Long-term Relevance and Scaling with Stronger VLMs**
> We thank the reviewer for suggesting using stronger VLMs. To fulfill the reviewer's requests, we have conducted additional qualitative experiments using Gemini on examples from ALFWorld and R2R https://shorturl.at/3jJiQ. This examples reveals that raw perception often yields overly general descriptions, while goal-aware perception can suffer from hallucinations, such as mentioning a CD not present in the image.
>
> Moreover, as shown on **Appendix B.5.1**, we evaluate our framework using latest open-source models **InternVL3 (released in 2025)**. We observed that even with these recent backbones, "Raw Perception" often produces descriptions that lack the specific details required for the task. Applying PRISM to InternVL3 consistently improved the relevance and quality of the descriptions. This confirms that the "perception-decision gap" is a structural issue that better base perception alone cannot solve.
>
> # **Conclusion:**
> We thank the reviewer again for these valuable suggestions. These clarifications regarding latency, the significance testing, scaling with stronger VLMs, and the detailed error analysis have significantly strengthened the paper and will be included in the final version.

---

> > ### Author Rebuttal · Reviewer_Lz6B · 2026-04-05
> >
> > I will maintain my positive score.

---

### Official Review · Reviewer_g6Mg · 2026-03-13

**Soundness:** 3
**Presentation:** 3
**Significance:** 3
**Originality:** 3
**Overall Recommendation:** 5
**Confidence:** 3

**Summary:**

The paper proposes PRISM (Perception and Reasoning Interleaved for Sequential Decision Making), a modular framework designed to bridge the perception-reasoning gap in embodied agents. Unlike traditional decoupled systems, PRISM introduces a Dynamic Question-Answering (DQA) pipeline where a Large Language Model (LLM) actively critiques initial Vision-Language Model (VLM) descriptions and queries the VLM for task-critical information. The agent then synthesizes these interactions into a concise, task-driven scene representation for final action selection.

**Compliance With Llm Reviewing Policy:**

Affirmed.

**Final Justification:**

The author’s rebuttal has resolved my concern about inference efficiency and the errors in VLM perception. I changed my rating from 4 weak accept to 5 accept.

**Key Questions For Authors:**

Refer to the weakness part.

**Limitations:**

Refer to the weakness part.

**Strengths And Weaknesses:**

- Strengths:
    - Effective Coupling: The DQA mechanism successfully creates a closed-loop interaction between perception and reasoning, ensuring descriptions prioritize goal-relevant details.
    - Performance and Robustness: The framework achieves substantial gains in success rates (SR) across multiple benchmarks, narrowing the gap between visual and perfect-text perception.
    - Concise Synthesis: By merging DQA feedback into a single description rather than using raw concatenation, the model reduces token noise and improves decision-making efficiency.
- Weaknesses:
    - Inference Latency: The multi-turn interaction between VLM and LLM at each time step likely increases computational overhead and latency compared to monolithic or single-pass models. Could you provide a quantitative analysis of the inference time and token consumption compared to standard modular baselines (e.g., VIPER or DiscussNAV)?
    - Potential Error Propagation: The framework’s performance remains dependent on the initial VLM's capacity; incorrect VLM answers to DQA queries may lead to hallucinated or erroneous task-driven descriptions. How does the system handle conflicting information where the VLM's answer contradicts its initial description or the agent's prior history?
    - Scalability to Complex Dynamics: While effective in simulated environments, the robustness of the iterative DQA process in highly dynamic or visually noisy real-world settings is not fully explored.

---

> ### Author Rebuttal · Authors · 2026-03-27
>
> # **Reviewer Response**
> We sincerely thank the reviewer for the positive assessment of PRISM. We are encouraged that the reviewer recognized the "effective coupling" of our DQA mechanism, the **substantial gains in success rates** narrowing the gap with perfect-text perception, and the **concise synthesis** which improves decision-making efficiency. Below, we address the specific concerns raised in **Weaknesses** (W1 to W3) regarding latency, perception robustness and scalability to complex dynamics.
> ## **W1. Inference Latency and Operational Efficiency:**
> To address the concern regarding computational cost, we provide a systematic analysis comparing PRISM (using Idefics2 / Qwen3 4B) against the Raw Perception (No QA) baseline. All experiments were conducted on an NVIDIA H100 GPU.
> We emphasize that **PRISM operates at the high-level decision-making phase.** At this stage, the agent identifies the correct high-level action required to progress toward the goal. In such a context, **decision latency is significantly less critical than the accuracy and efficiency of the physical execution** that follows.
> **Results:**
> | Metric | Raw Perception (No QA) | PRISM (Ours) |
> | :--- | :---: | :---: |
> | **Success Rate (SR)** | 69% | 75% |
> | **Avg. Steps Executed** | 20.9 | 17.1 |
> | **Inference Latency ($s/step$)** | 2.14s | 6.35s |
>
> **Analysis of Decision Quality vs. Execution Speed**
> The data illustrates that while PRISM requires about 3 times more time per individual step, corresponding to 3 forward passes (to determine the questions, to determine the answers, and to generate the updated description), it significantly optimizes the overall task execution:
> * Step Efficiency: **PRISM executes 18% fewer high-level actions in the environment (17.1 vs. 20.9).** This indicates that the interactive QA process allows the agent to select more direct and effective actions.
> * Reduction of Irrelevant Decisions: **The additional "perception refinement time" (6.35s vs. 2.14s) prevents the agent from making irrelevant or redundant moves.** By reducing VLM error through targeted questioning, we ensure each high-level action is purposeful.
> * Performance Gain: **Prioritizing the quality of each choice results in a 6 pt absolute gain in Success Rate.** In complex tasks, **taking fewer, higher-quality actions is far more sustainable and successful** than taking many rapid, inaccurate ones.
> ## **W2. Robustness to VLM Perception Errors**
> The reviewer’s question provides us the opportunity to clarify how PRISM handles potential VLM errors. We already show
> in **Table 4** of the paper that our automated pipeline achieves Success Rates comparable to both Oracle Q and Oracle QA, suggesting that **perception errors do not significantly degrade overall performance.**
> To further quantify this, we have performed additional experiments to evaluate VLM (Idefics2) performance across 1,000 transitions by comparing its outputs against ground-truth answers extracted directly from the text-based environment to isolate perception accuracy. We get the following results:
> **VLM Performance Results:**
> | Metric | Pick | Look | Clean | Heat | Cool | Pick2 | **Avg** |
> | --- | --- | --- | --- | --- | --- | --- | --- |
> | **Precision** | 0.85 | 0.91 | 0.96 | 0.96 | 0.83 | 0.95 | **0.91** |
> | **Recall** | 0.89 | 0.90 | 0.95 | 1.00 | 0.87 | 0.97 | **0.93** |
> | **F1** | 0.87 | 0.90 | 0.95 | 0.98 | 0.85 | 0.96 | **0.92** |
> **Analysis of Results:**
> * Corroboration of Table 4: **The average F1-score of 0.92 explains why Success Rates remain stable in Table 4; the minor perception errors (averaging 8%) are typically non-critical and do not lead to catastrophic error propagation.**
> * Accuracy and Reliability: **A 0.91 Precision indicates a low VLM error rate, while 0.93 Recall confirms the model consistently identifies relevant information.**
> * Failure Modes: Performance is highest in categories like Heat and Clean. The lower metrics in Cool and Pick typically stem from visual complexity in the environment.
> ## **W3 – Scalability to Complex Dynamics**
> We thank the reviewer for raising this point. Our evaluations are conducted on two environments (ALFWorld and Room-to-Room), with R2R offering a more realistic visual setting. As mentioned in the conclusion, our future work involves extending PRISM to low‑level control in robotics, which will entail deploying the iterative DQA pipeline on physical robots operating under dynamic, visually noisy real‑world conditions.
>
> # **Conclusion**
> We thank the reviewer again for these valuable suggestions. This feedback and the resulting clarifications have significantly strengthened the paper, we will ensure these analyses and specifications are explicitly included in the final version.

---

> > ### Author Rebuttal · Reviewer_g6Mg · 2026-04-02
> >
> > Thanks for the rebuttal. My concerns have been resolved, and I will raise the rating to 5.

---

### Official Review · Reviewer_bZL8 · 2026-03-16

**Soundness:** 3
**Presentation:** 3
**Significance:** 3
**Originality:** 3
**Overall Recommendation:** 4
**Confidence:** 3

**Summary:**

This paper proposes PRISM, a framework that interleaves perception and reasoning for sequential decision making in embodied environments. Instead of relying on a single-pass visual description, PRISM allows an LLM to iteratively generate task-relevant questions about the visual observation, which are answered by a VLM to gather missing information. The initial scene description and the question–answer results are then synthesized into a refined textual state representation used by a policy model to predict actions. Experiments on ALFWorld and Room-to-Room (R2R) show that PRISM significantly improves success rates over monolithic VLM agents and standard modular pipelines. The results suggest that goal-driven interactive perception helps reduce irrelevant descriptions and improves downstream decision making.

**Compliance With Llm Reviewing Policy:**

Affirmed.

**Key Questions For Authors:**

1. Could the authors provide an analysis of the additional computational cost and latency compared to standard modular pipelines?
2. Since the final scene description relies on VLM-generated answers, how robust is the framework to perception errors such as hallucinated objects or incorrect QA responses? Have the authors analyzed how often such errors propagate to the decision stage?
3. How sensitive is the performance to the number of generated questions, and is there an optimal trade-off between performance and computational efficiency?

**Limitations:**

Yes

**Strengths And Weaknesses:**

# Strengths
- This paper introduces an intuitive pipeline where perception and reasoning are interleaved through LLM-generated questions, enabling the agent to actively gather task-relevant visual information instead of relying on static captions.
- This paper clearly identifies limitations of both end-to-end VLM agents and traditional modular pipelines, motivating the need for task-driven perception.
- This paper is well-structured and easy to understand for the audience.

# Weakness
1. The interactive QA process introduces multiple inference steps (captioning, question generation, answering, synthesis), which may increase latency and make real-time deployment challenging. It would be helpful to quantify the extra computational cost or inference time incurred by these steps.
2. Although the interactive QA process is intended to improve perception quality, errors from the VLM (e.g., hallucinated objects or incorrect answers) may still propagate into the synthesized scene description. The paper does not provide a systematic analysis of how often such perception errors occur or how robust the system is to them.
3. The framework relies on generating multiple questions to refine scene understanding, but the paper provides limited analysis on how the number of questions affects performance, efficiency, and robustness.

---

> ### Author Rebuttal · Authors · 2026-03-27
>
> # **Reviewer Response**
> We sincerely thank the reviewer for their insightful and constructive feedback. We are encouraged that the reviewer found our pipeline **intuitive**, recognized the motivation behind **task-driven perception,** and noted that the paper is **well-structured and easy to understand.** Below, we address the specific concerns raised in “Weaknesses” (W1 to W3) regarding latency, perception robustness and the impact of the number of questions.
> ## **W1. Inference Latency and Operational Efficiency:**
> To address the concern regarding computational cost, we provide a systematic analysis comparing PRISM (using Idefics2 / Qwen3 4B) against the Raw Perception (No QA) baseline. All experiments were conducted on an NVIDIA H100 GPU.
> We emphasize that **PRISM operates at the high-level decision-making phase.** At this stage, the agent identifies the correct high-level action required to progress toward the goal. In such a context, **decision latency is significantly less critical than the accuracy and efficiency of the physical execution** that follows.
> **Results:**
> | Metric | Raw Perception (No QA) | PRISM (Ours) |
> | --- | --- | --- |
> | **Success Rate** | 69% | 75% |
> | **Avg. Steps Executed** | 20.9 | 17.1 |
> | **Inference Latency ($s/step$)** | 2.14s | 6.35s |
>
> **Analysis of Decision Quality vs. Execution Speed**
> The data shows that while PRISM requires about 3 times more time per individual step, corresponding to 3 forward passes (to determine the questions, to determine the answers, and to generate the updated description), it significantly optimizes the overall task execution:
> * Step Efficiency: **PRISM executes 18% fewer high-level actions in the environment (17.1 vs. 20.9).** This indicates that the interactive QA process allows the agent to select effective actions.
> * Reduction of Irrelevant Decisions: **The additional "perception refinement time" (6.35s vs. 2.14s) prevents the agent from making irrelevant or redundant moves.** By reducing VLM error through targeted questioning, we ensure each high-level action is purposeful.
> * Performance Gain: **Prioritizing the quality of each choice results in a 6 pt absolute gain in Success Rate.** In complex tasks, **taking fewer, higher-quality actions is far more sustainable and successful** than taking many rapid, inaccurate ones.
> ## **W2. Robustness to VLM Perception Errors and Hallucinations**
> The reviewer’s question provides us the opportunity to clarify how PRISM handles potential VLM errors. We already show
> in **Table 4** of the paper that our automated pipeline achieves Success Rates comparable to both Oracle Q and Oracle QA, suggesting that **perception errors do not significantly degrade overall performance.**
> To further quantify this, we have performed additional experiments to evaluate VLM (Idefics2) performance across 1,000 transitions by comparing its outputs against ground-truth answers extracted directly from the text-based environment to isolate perception accuracy. We get the following results:
> **VLM Performance Results:**
> | Metric | Pick | Look | Clean | Heat | Cool | Pick2 | **Avg** |
> | --- | --- | --- | --- | --- | --- | --- | --- |
> | **Precision** | 0.85 | 0.91 | 0.96 | 0.96 | 0.83 | 0.95 | **0.91** |
> | **Recall** | 0.89 | 0.90 | 0.95 | 1.00 | 0.87 | 0.97 | **0.93** |
> | **F1** | 0.87 | 0.90 | 0.95 | 0.98 | 0.85 | 0.96 | **0.92** |
> **Analysis of Results:**
> * Corroboration of Table 4: **The average F1-score of 0.92 explains why Success Rates remain stable in Table 4; the minor perception errors (averaging 8%) are typically non-critical and do not lead to catastrophic error propagation.**
> * Accuracy and Reliability: **A 0.91 Precision indicates a low VLM error rate, while 0.93 Recall confirms the model consistently identifies relevant information.**
> * Failure Modes: Performance is highest in categories like Heat and Clean. The lower metrics in Cool and Pick typically stem from visual complexity in the environment.
> ## **W3. Impact of the Number of Questions on Performance**
> We thank the reviewer for highlighting the need for a sensitivity analysis regarding the interaction budget. We have detailed this analysis in **Appendix B.5.2**:
> * Sensitivity Analysis: We evaluated the impact of varying the question budget (e.g., limiting the model to 1 or 3 questions).
> * Comparative Results: **As shown in Table 9, PRISM achieves its best and most stable results when given a 3-question limit.** Notably, our automatic pipeline yields similar performance results to the 3-question limit.
> * Autonomous Adaptation (Figure 7): We conducted an analysis of the number of questions generated by PRISM under autonomous adaptation, as shown in **Figure 7**. We found that **PRISM dynamically adapts the number of questions based on the specific task and scene complexity.**
>
> # **Conclusion:**
> We thank the reviewer again for these valuable suggestions. This feedback and the resulting clarifications will help us strengthen the final version of the paper.

---

> > ### Author Rebuttal · Reviewer_bZL8 · 2026-04-04
> >
> > Thank you for the thoughtful rebuttal. The added clarifications improve the paper, and I appreciate the authors’ efforts to address the feedback. I will keep my evaluation unchanged.

---

### Decision · Program_Chairs · 2026-04-30

**Decision:**

Accept (regular)

**Comment:**

This paper proposes PRISM, a framework that interleaves perception and reasoning via a dynamic question-answering mechanism to improve embodied decision-making. All reviewers agree that the paper is well-written, technically sound, and demonstrates consistent empirical gains across benchmarks. The proposed DQA mechanism is considered effective in bridging the perception–decision gap and improving task-relevant scene understanding.

The main concerns raised during the review process focus on computational efficiency, potential error propagation from VLM outputs, and limited analysis of statistical robustness and scalability. However, the authors have provided a thorough rebuttal addressing these issues, including quantitative latency analysis, perception error evaluation, and additional statistical validation. Reviewers acknowledged that these concerns were adequately resolved and maintained or increased their scores accordingly.

Overall, given the solid methodology, clear empirical improvements, and satisfactory rebuttal, I recommend accepting this paper.